# Inorganic nitrogen inhibits symbiotic nitrogen fixation through blocking NRAMP2-mediated iron delivery in soybean nodules

Min Zhou[1,5], Yuan Li[1,5], Xiao-Lei Yao[1,5], Jing Zhang [1], Sheng Liu[1], Hong-Rui Cao[1], Shuang Bai[1], Chun-Qu Chen[1], Dan-Xun Zhang[1], Ao Xu[1], Jia-Ning Lei[2], Qian-Zhuo Mao[2], Yu Zhou[3], De-Qiang Duanmu [3] ✉, Yue-Feng Guan [4] ✉ & Zhi-Chang Chen [1] ✉

Symbiotic nitrogen fixation (SNF) in legume-rhizobia serves as a sustainable source of nitrogen (N) in agriculture. However, the addition of inorganic N fertilizers significantly inhibits SNF, and the underlying mechanisms remain not-well understood. Here, we report that inorganic N disrupts iron (Fe) homeostasis in soybean nodules, leading to a decrease in SNF efficiency. This disruption is attributed to the inhibition of the Fe transporter genes *Natural Resistance-Associated Macrophage Protein 2a and 2b* (*GmNRAMP2a&2b)* by inorganic N. GmNRAMP2a&2b are predominantly localized at the tonoplast of uninfected nodule tissues, affecting Fe transfer to infected cells and consequently, modulating SNF efficiency. In addition, we identified a pair of N-signal regulators, nitrogen-regulated GARP-type transcription factors 1a and 1b (GmNIGT1a&1b), that negatively regulate the expression of *GmNRAMP2a&2b*, which establishes a link between N signaling and Fe homeostasis in nodules. Our findings reveal a plausible mechanism by which soybean adjusts SNF efficiency through Fe allocation in response to fluctuating inorganic N conditions, offering valuable insights for optimizing N and Fe management in legume-based agricultural systems.

Legumes have a natural ability to fix atmospheric nitrogen (N₂) into organic form through their N-fixing symbiosis system, making them a major sustainable source of nitrogen for agriculture[1]. However, despite providing ~50 million tons of N per annum into agricultural systems, symbiotic nitrogen fixation (SNF) by legumes still falls short of the amount provided by inorganic N fertilizers[2]. Excessive use of N fertilizers not only incurs environmental and economic costs, but also inhibits SNF in legumes[3]. Understanding the mechanisms underlying this inhibition can help develop strategies to balance nodule-based N fixation with soil N fertilization.

The nutrient exchange between legumes and rhizobia is a mutually beneficial and essential aspect of their symbiosis, where rhizobia provide fixed N to legumes in exchange for carbohydrates and mineral elements. This exchange takes place in root nodules of legumes[4]. One of the mineral elements provided by host plants is iron (Fe), which is indispensable for SNF in rhizobia[5]. Root nodules of

[1]Haixia Institute of Science and Technology, Fujian Agriculture and Forestry University, Fuzhou, China. [2]State Key Laboratory for Managing Biotic and Chemical Threats to the Quality and Safety of Agro-products, Key Laboratory of Biotechnology in Plant Protection of MARA, Key Laboratory of Green Plant Protection of Zhejiang Province, Institute of Plant Virology, Ningbo University, Ningbo, China. [3]State Key Laboratory of Agricultural Microbiology, Hubei Hongshan Laboratory, Huazhong Agricultural University, Wuhan, China. [4]Guangdong Provincial Key Laboratory of Plant Adaptation and Molecular Design, Innovative Center of Molecular Genetics and Evolution, School of Life Sciences, Guangzhou University, Guangzhou, Guangdong, China. [5]These authors contributed equally: Min Zhou, Yuan Li, Xiao-Lei Yao. ✉e-mail: duanmu@mail.hzau.edu.cn; guan@gzhu.edu.cn; zcchen@fafu.edu.cn

legumes typically contain a higher concentration of Fe than other vegetative organs. At the maturity stage, more than 40% of Fe accumulates in soybean nodules[6]. The large amount of Fe is used as cofactors and components of the proteins (nitrogenase, leghemoglobin, ferredoxin, etc.) that are essential for SNF[5]. However, Fe deficiency can severely inhibit nodule formation and development, and consequently, SNF efficiency[7,8], especially for legumes grown in calcareous soils where Fe becomes poorly soluble[9].

Vacuolar Fe plays an important role in maintaining Fe homeostasis in plants. Plants can sequester excess Fe to the vacuoles, preventing its toxic accumulation. When Fe is deficient, vacuolar Fe can be mobilized and transported to other parts of the plant, providing a readily available source of Fe for metabolic processes[10,11]. This movement of Fe is often mediated by Fe transport proteins at the tonoplast. In Arabidopsis, Fe can be compartmentalized in vacuoles of embryos by VIT1 and mobilized by AtNRAMP3&4 during seed germination[12]. Similarly, legume nodules rely on a complex system of Fe transporters to maintain Fe homeostasis, with numerous transport family members, including NRAMP, VIT, YSL, ZIP, and MATE, working together to move Fe from the vasculature to the infected cells, and finally to the basic nitrogen-fixing unit, symbiosome[1,5,13]. Despite the advancements made in nodule Fe transport, it remains unclear if nodules have specialized Fe-storing cells and relevant Fe transporters.

Legumes have an auto-regulation of nodulation (AON) system that responds to external N sources and fine-tunes nodulation to prevent carbon loss when N is abundant[1]. During the stages of SNF, metabolic and transport processes associated with SNF are precisely regulated to adapt to external N[14]. High levels of inorganic N can cause premature aging of the nodules, thus terminating the transfer of nutrients from the host plant to the nodules[15]. Transcription factors, Nodule Inception (NIN), and NIN-like proteins (NLP) play crucial roles in both nodulation and SNF in legumes[1,16,17]. These N-responsive NIN / NLPs modulate signaling pathways responsible for nodule formation and development[18,19]. Recent studies have found that NLP2 and NIN can directly activate the expression of leghemoglobin, an oxygen-binding phytoglobin that carries heme (an Fe-containing molecule) in nodules of *Medicago truncatula*[16]. However, the mechanism by which inorganic N regulates nodule Fe homeostasis is still largely unknown.

In this study, we found that the addition of inorganic N significantly disrupts the Fe homeostasis in soybean nodules, thereby affecting SNF. This process is attributed to the inhibition of the expression of the Fe transporter genes *GmNRAMP2a&2b* by inorganic N. A previous report suggested that GmNRAMP2b (also known as GmDMT1) facilitated Fe transport from the cytoplasm of infected cells into symbiosomes[20], functioning in a similar manner to VIT-like (VTL) transporters[21–23]. However, our findings reveal that both GmNRAMP2a&2b were Fe influx transporters localized at the tonoplast of uninfected tissues of nodule. They act in a manner of genetic compensation, to facilitate Fe transfer from uninfected to infected cells, which are indispensable for SNF in soybean nodules. We further demonstrate that the expression of *GmNRAMP2a&2b* is negatively regulated by two Nitrate-Inducible GARP-type Transcriptional Repressor (NIGT) family members, which reveals a regulatory network for N-dependent fine-tuning of Fe transport in symbiotic systems.

## Results

### Inorganic N inhibits SNF by disrupting Fe homeostasis

To investigate the effects of inorganic N on nodule Fe homeostasis, we conducted Perls/DAB staining on nodules exposed to high levels of inorganic N (H−N). Our observations revealed that Fe accumulated predominantly in the nitrogen fixation zones of the nodules. Furthermore, with prolonged N supply, the Fe signal gradually diminished (Fig. 1a, b), which was also accompanied by a reduction in N export rate in nodules (Fig. 1c). Consistently, Fe content in symbiosome was gradually decreased after transfer to H−N (Fig. 1d), suggesting that

inorganic N blocks Fe delivery and disrupts Fe balance within the nodules. Subsequently, we assessed the inhibitory effects of N on SNF under varying Fe concentrations. By analyzing nodule N export rates and the proportion of RFP-expressing rhizobia within infected cells, we observed that high Fe supply mitigated the inhibitory effects of N on SNF, while low Fe resulted in the opposite outcome (Fig. 1e–g). These findings collectively suggest that disruption of Fe homeostasis may be one of the pathways through which inorganic N inhibits SNF in nodules.

### Expression of *GmNRAMP2a&2b* in nodules is responsive to external N and Fe levels

To investigate the molecular basis on N-regulated Fe homeostasis in nodules, we first carried out comparative RNA-seq analysis of mature nodules and screened out 142 genes responding to both Fe deficiency (-Fe) and H-N (Fig. 2a, Supplementary Data 1). Among these genes, we classified Fe homeostasis-related genes into four clusters based on their response patterns (Fig. 2b, c). Intriguingly, two *NRAMP* genes were both down-regulated by H-N and up-regulated by -Fe (Fig. 2c). A phylogenetic tree of NRAMPs from rice, soybean and Arabidopsis was constructed and the result suggested that these two *NRAMPs* are the most similar homologues in the soybean genome (Supplementary Fig. 1a). They are 98% identical at the amino acid level and very conserved in their trans-membrane domains (Supplementary Fig. 1b), suggesting that they may play complementary roles in their biological functions. Combined with the previous report that they are a pair of paralogs resulting from genome duplication events[24], we therefore named them *GmNRAMP2a* and *GmNRAMP2b* hereinafter.

*GmNRAMP2a* and *2b* are expressed in most tissues according to the Phytozome database, and only *GmNRAMP2b* is highly expressed in nodules (Fig. 2d). The expression of these two genes in other tissues may play a role similar to that of AtNRAMP3 and 4 in Arabidopsis[25]. Real-time RT-PCR showed that *GmNRAMP2b* was primarily expressed in the fixation zone of nodules (Fig. 2e). *GmNRAMP2b* exhibited much higher expression levels than *GmNRAMP2a* (Fig. 2e). The expression of *GmNRAMP2b* but not *GmNRAMP2a* was gradually increased with the days after inoculation, reaching its peak at 21 days and subsequently declining gradually (Fig. 2f). Consistent with RNA-seq data, both genes were up-regulated by -Fe and down-regulated by H−N, but were not varied by other nutrient deficiencies, such as Mg, Mo, S, Mn or Zn (Fig. 2g). Notably, under both -Fe and H−N conditions, the expression of *GmNRAMP2a* and *2b* remained suppressed (Fig. 2h, i), suggesting that N signals play a dominant role in expression regulation. Furthermore, regardless of whether the N source is ammonium or nitrate, both can trigger the suppression of gene expression (Supplementary Fig. 2a, b).

### GmNRAMP2a&2b are tonoplast-targeted proteins in nodule uninfected tissues

To verify their tissue and cell specificities, we generated transgenic hairy roots carrying green fluorescent protein (GFP) driven by *GmNRAMP2a* or *2b* promoters. In situ immunostaining results from nodules at 17 and 30 dpi showed that GFP antibody (anti-GFP) signals were observed mainly in fixation zone of *pGmNRAMP2a:GFP* transgenic nodules under Fe-deficient conditions, whereas no signals could be observed under Fe-sufficient conditions (Fig. 3a, Supplementary Fig. 3a). By contrast, anti-GFP signals can be detected in *pGmNRAMP2b:GFP* transgenic nodules under both Fe-deficient and -sufficient conditions (Fig. 3b, Supplementary Fig. 3b). Using an RFP-expressing rhizobium strain as a marker for infected cells, we found that both genes were expressed in those cells that were not colonized by rhizobia (Fig. 3c, d, Supplementary Fig. 3c, d), indicating the uninfected cell-specific expression of *GmNRAMP2a&2b*. The signal intensity from both transgenic nodules showed a -Fe-induced pattern, and followed the order: Fixation zone > Vasculature > Cortex (Fig. 3e,

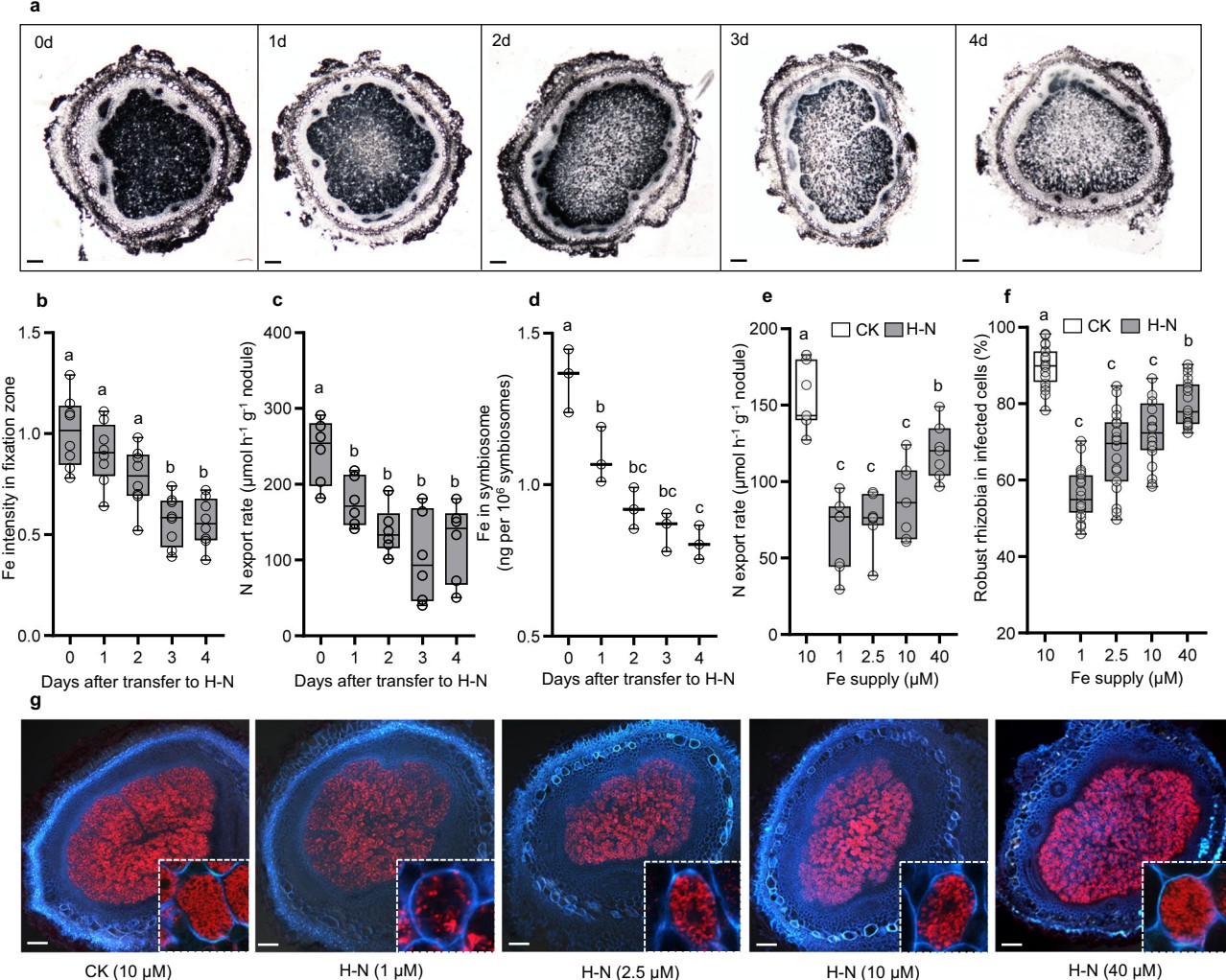

**Fig. 1 | Inorganic N disrupts Fe homeostasis and inhibits SNF in nodules. a–d** Fe accumulation and SNF affected by inorganic N in nodules. Nodules at 21 dpi were transplanted to a high-N nutrient solution (H–N) for 0, 1, 2, 3, 4 d, and related parameters were determined, including (**a**) Fe distribution in nodules by perls/DAB staining. 8 independent nodule sections were investigated and consistent results were obtained, with one representative image presented in (**a**). Scale bars, 100 μm. **b** Signal intensity of perls-DAB staining in fixation zone. **c** Ureide export rate of nodules. **d** Fe content in symbiosome of nodules. **e–g** High Fe supply mitigated the inhibitory effects of inorganic N on SNF. Nodules at 7 dpi were grown in various Fe conditions for 12 days, and then treated with H–N for 2 d. Related parameters were determined, including (**e**) Ureide export rate of nodules. **f, g** Occupation of RFP-expressing rhizobia in infected cells. Cyan shows signals from cell wall; Red shows RFP-tagged rhizobia. Images in white dot lines are magnified infected cells. 20 independent nodule sections were investigated and consistent results were obtained, with one representative image presented. Scale bars, 100 μm. The numbers in parentheses in (**g**) represent the concentration of Fe supply. The boxes in (**b-f**) indicate the first and third quartiles, and the whiskers indicate the minimum and maximum values. The lines within the boxes indicate the median values. $n = 8$ (**b**), 6 (**c**), 7(**e**), 20 (**f**) biologically independent replicates, or 3 (**d**) independent pools (1 g of nodules per pool). The different letters in (**b-f**) indicate significant differences (adjusted $P \leq 0.05$) in multiple comparisons tests following two-sided Tukey tests. Source data are provided as a Source Data file.

Supplementary Fig. 3e). Furthermore, Fe limitation did not alter uninfected cell-specific expression of *GmNRAMP2b* (Fig. 3f, Supplementary Fig. 3f).

We next examined the subcellular localization of GmNRAMP2a&2b in tobacco (*Nicotiana tabacum*) leaf protoplasts. Fluorescence signals from both GmNRAMP2a-GFP and GmNRAMP2b-GFP proteins were observed at the tonoplast, which were easily distinguishable from signals of the plasma membrane (PM) marker FM4-64 FX and chlorophyll autofluorescence (Fig. 3g, h). We subsequently investigated the subcellular localizations of GmNRAMP2a&2b in yeast, and found both of them localized specifically to vacuolar membrane (Supplementary Fig. 4a). To further verify their tissue and subcellular localizations in soybean nodules, we generated transgenic hairy roots carrying *pGmNRAMP2a/2b: GmNRAMP2a/2b-GFP*, and inoculated them with rhizobium. To obtain a more clearly visible signal, we treated

*pGmNRAMP2a: GmNRAMP2a-GFP* transgenic nodules with Fe deficiency. In situ immunostaining results from nodules at 17 and 30 dpi showed that in both transgenic nodules, anti-GFP signals mainly located in uninfected cells (smaller cell size) of fixation zone that was in close proximity to infected cells (larger cell size; Fig. 3i–j, Supplementary Fig. 3g-h). They were also observed in the pericycle of nodule vascular tissues (Supplementary Fig. 5a, b). Furthermore, these signals showed a ring-like structure inside the cell but outside the nucleus (Fig. 3i–j, Supplementary Figs. 3g, h, 5c, d), suggesting that GmNRAMP2a&2b are targeted to the cell tonoplast. We next examined their protein levels under H–N supply, and found that both proteins (GmNRAMP2b under +Fe and GmRNAMP2a under -Fe conditions) were very susceptible to H–N supply, with a rapid decrease of anti-GFP signal abundances in nodules after being transferred to H–N for 12 and 24 h (Fig. 3k–l, Supplementary Figs. 5e, f).

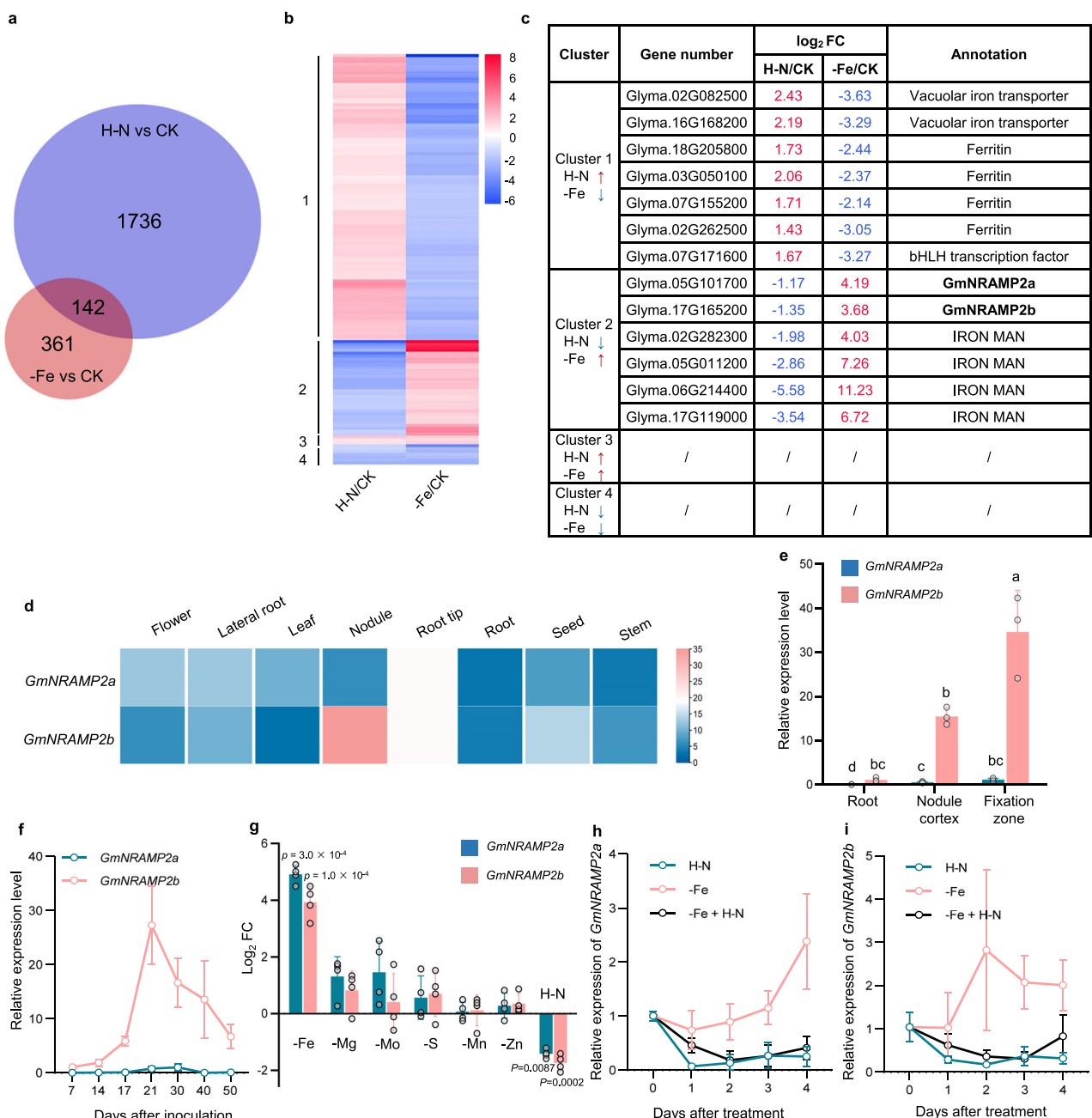

**Fig. 2 | Fe- and N-responsive expression of *GmNRAMP2a&2b* in nodules.**
**a–c** Screening of Fe- and N-responsive genes by RNA-seq. Nodules collected at 17 dpi, which experienced either 7 days of Fe deficiency (-Fe) or 1 day of high-N (H–N), were used for RNA-seq analysis. Low-N and +Fe treatments were used as CK. **a** Venn diagram showing number of -Fe and H–N regulated genes. **b** Heatmap of 142 differentially expressed genes. Colors represent Log$_2$ FC. FC, fold change. **c** List of four classified clusters showing Fe homeostasis related genes. **d–i** Gene expression pattern of *GmNRAMP2a&2b*, including (**d**) Transcriptional abundance in tissues/organs. Data are from Phytozome and FPKM (Fragments Per Kilobase of transcript per Million mapped reads) values are shown. **e** Gene expression in nodules. Nodules at 21 dpi were separated into three parts for RNA extraction: nodule conjugated root segments with nodules removed, nodule cortex, and fixation zone. **f** Time-

dependent expression in nodules. 4-d-old seedlings were inoculated with rhizobia and cultured in low-N solution for different days. **g** The expression response to nutrient stresses. Nodules collected at 17 dpi, which experienced either 7 days of metal deficiencies, or 1 day of H–N, were used for RNA extraction. **h, i** The expression response to N and Fe interaction. Nodules at 21 dpi were treated with H–N, -Fe or a combination of both for different days. Relative expression levels were determined by real-time RT-PCR. *EF-1α* was used as an internal standard. Data are means ± SD. $n = 3$ (**e, f, h, i**) or 4 (**g**) biologically independent replicates. The different letters in (**e**) indicate significant differences (adjusted $P \le 0.05$) in multiple comparisons tests following two-sided Tukey tests. Values are mean ± SD. Two-tailed t-test (**g**). Source data are provided as a Source Data file.

GmNRAMP2b was previously identified on the symbiosome membrane (SM) via immunoelectron microscopy[20]. We also used this method with GFP-specific antibodies for confirmation. However, we found that GmNRAMP2b-GFP was not detected on the SM but specifically located on the tonoplast of uninfected cells (Supplementary Fig. 6a). This subcellular localization was further confirmed by western

blot analysis, which showed that the GmNRAMP2b protein tagged with GFP displayed the same fractionation pattern as the V-type ATPase (a known tonoplast membrane marker protein), but differed from the pattern shown by H$^+$-ATPase (a known plasma membrane marker protein). Furthermore, we did not detect the GFP-tagged GmNRAMP2b protein in any subcellular structure of the symbiosomes, whereas the

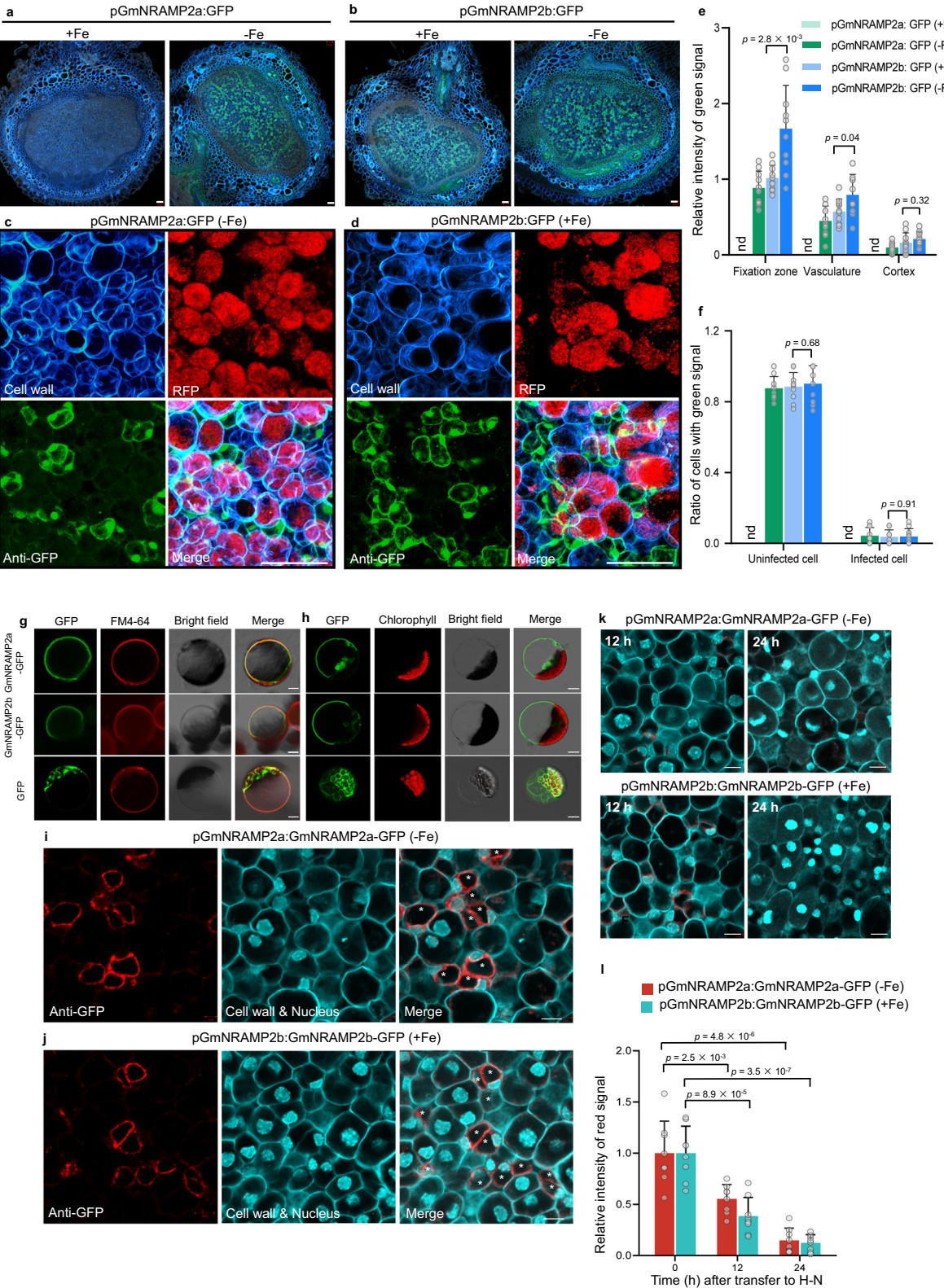

Nodulin-26 protein (a marker protein for the SM) showed a specific localization on the SM (Supplementary Fig. 6b). These results indicate that GmNRAMP2b has a specific localization to the tonoplast of uninfected cells.

## Yeast complementation test of GmNRAMP2a&2b

To investigate the Fe transport activity, we transformed the ORFs of *GmNRAMP2a*, *GmNRAMP2b*, *GmVTL1a*[21], and the full-length (FL) cDNA

of *GmNRAMP2b* individually into yeast WT strain BY4741 or *Δccc1* mutant[26], and then isolated vacuoles for Fe determination. The purity of extracted vacuoles was qualified by vacuolar marker ALP (Fig. 4a) and integrity was examined by FM4-64 staining (Supplementary Fig. 4b). We found the vacuolar Fe content in *GmVTL1a* transformants was remarkably higher than vector control after gene induction by galactose for 1 or 2 h. In contrast, the Fe content in *GmNRAMP2a&2b* and *GmNRAMP2b-FL* transformants was much lower than vector

**Fig. 3 | GmNRAMP2a&2b are located at the tonoplast of uninfected cells.**
**a–f** Immunostaining of transgenic nodules expressing *pGmNRAMP2a: GFP* (**a, c**) and
*pGmNRAMP2b: GFP* (**b, d**). Magnified images in fixation zone are shown in (**c, d**),
with individual channels and overlay (merge). Transgenic nodules at 10 dpi from
hairy roots were transplanted to Fe-free (-Fe) or Fe-sufficient ( + Fe) solutions for 7
d. Cyan shows signals from cell wall; Red shows RFP-tagged rhizobia; Green shows
anti-GFP signals. Five independent transgenic lines were investigated and con-
sistent results were obtained, with one representative image presented in (**a–d**).
Scale bars, 50 μm. **e** Relative intensity of green signal from (**a, b**). **f** Ratio of cells with
green signal in (**c, d**). nd means non-detected values. **g, h** Tobacco leaf protoplasts
expressing *GmNRAMP2a/2b-GFP* or *GFP* alone. Green shows GFP signals; red shows
PM marker (FM4-64 FX) signals or chlorophyll autofluorescence; gray shows bright

field; merged images show the combined three channels. **i, j** Immunostaining
staining of transgenic nodules expressing *pGmNRAMP2a/2b:GmNRAMP2a/2b-GFP*.
The uninfected cells were asterisk-marked in merged images. **k, l** Protein abun-
dance response to N. Cyan shows signals from cell wall and nucleus. Red shows anti-
GFP signals. Transgenic nodules at 10 dpi from hairy roots were transplanted to Fe-
free (-Fe) or Fe-sufficient ( + Fe) solutions for 7 d, and then treated with high-N
(H−N) for 0, 12 or 24 h. Five independent transgenic lines were investigated and
consistent results were obtained, with one representative image presented in (**g–k**).
Scale bars, 10 μm. Data are means + SD. *n* = 10 (**e, f**) or 8 (**l**) replicates from inde-
pendent nodules. The *P* values in (**e, f, l**) indicate significant differences compared
with +Fe or 0 h. Values are the mean ± SD (two-tailed *t*-test). Source data are pro-
vided as a Source Data file.

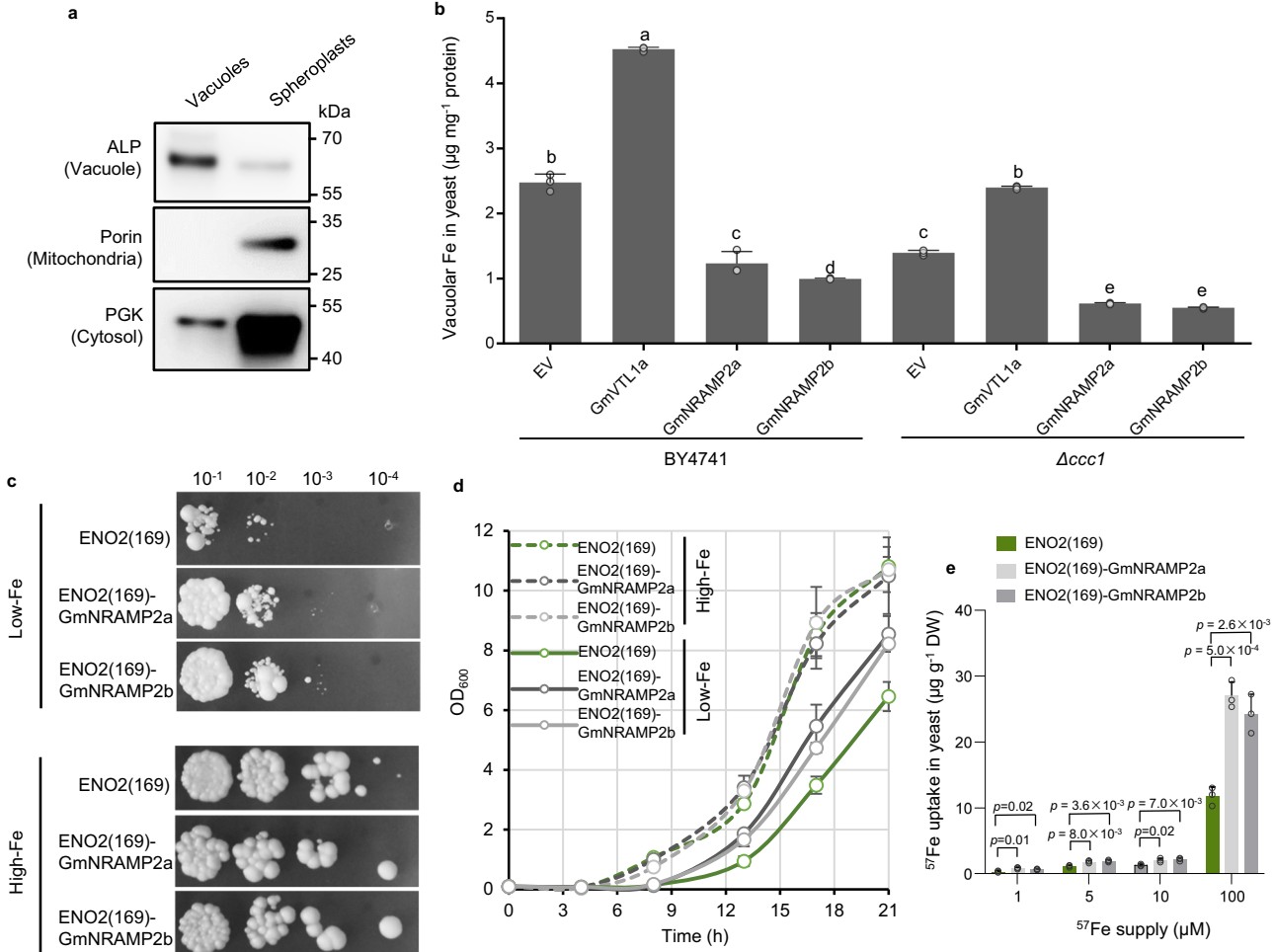

**Fig. 4 | Complementation test of GmNRAMP2a&2b in yeast. a** Examination of the
purity of vacuolar fraction by western blot analysis. Alkaline phosphatase (ALP) is a
vacuolar membrane protein; porin is a mitochondrial membrane protein; phos-
phoglycerate kinase (PGK) is a cytosolic protein. **b** Fe content in isolated yeast
vacuoles. BY4741 or *Δccc1* (mutant defective in vacuolar Fe storage) were trans-
formed with empty vector (EV), *GmVTL1a*, *GmNRAMP2a* or *2b*. Yeast vacuoles were
isolated by Ficoll gradient centrifugation, and used for Fe determination by ICP-MS.
**c** Yeast complementation assay. A yeast PM signal peptide ENO2(169) was fused to
GmNRAMP2a or 2b, and introduced into *fet3fet4* (mutant defective in Fe uptake).
Yeast transformants were grown on SD-uracil plates containing 20 (Low-Fe) or
100 μM (High-Fe) FeCl₃ for 3 d at 30 °C before being photographed. Four serial

dilutions of yeast cells starting from $OD_{600} = 0.1$ were spotted on plates.
**d** Proliferation of yeast cells. Yeast cell suspensions were diluted to an $OD_{600}$ of 0.1,
and then incubated in liquid medium with different concentrations of $FeCl_3$ at 30 °C
for 21 h. The values of $OD_{600}$ were determined. **e** Short-term $^{57}Fe$ uptake in yeast.
Yeast cells were collected after incubation with 1, 5, 10 or 100 μM $^{57}Fe$ for 5 min, and
then digested for $^{57}Fe$ determination by ICP-MS. Data are means + SD. *n* = 3 (**b, d, e**)
biologically independent replicates. The different letters in (**b**) indicate significant
differences (adjusted $P ≤ 0.05$) in multiple comparisons tests following two-sided
Tukey tests. The *P* values in (**e**) show significant differences compared with
ENO2(169). Values are the mean ± SD (two-tailed *t*-test). Source data are provided as
a Source Data file.

control in BY4741 strains (Supplemental Fig. 4c). Fe accumulation in
vacuoles was always lower in *Δccc1* mutant compared to BY4741, due
to defects in vacuolar Fe storage of *Δccc1* (Fig. 4b). However, *GmVTL1a*
and *GmNRAMP2a&2b* transformants in *Δccc1* mutant still exhibited the
same trends as observed in BY4741 (Fig. 4b). To further elucidate their

potential Fe transport abilities, we fused the PM signal peptide
ENO2(169)[27] in front of the ORF of *GmNRAMP2a* or *2b*, and transformed
them into *fet3fet4* strain. The ENO2(169)-fused GmNRAMP2a or 2b
showed both PM and tonoplast positioning (Supplementary Fig. 4d).
Although low-Fe supply makes *fet3fet4* strain grow poorly,

transformation with either *ENO2(169)-GmNRAMP2a* or *2b* promoted the growth of *fet3fet4* (Fig. 4c, d). In parallel, short-term $^{57}$Fe uptake experiment showed that Fe uptake in both *ENO2(169)-GmNRAMP2a&2b* transformants was significantly higher than the vector control (Fig. 4e). Taken together, these results indicate that GmNRAMP2a&2b are Fe influx transporters and mediate the transport of Fe from the vacuoles to the cytoplasm in yeast cells.

### GmNRAMP2a&2b modulate SNF activity by affecting Fe transfer to infected cells

Due to the dominant expression of *GmNRAMP2b* (Fig. 2e, f), we first generated a *nramp2b* single mutant by CRISPR-Cas9 for phenotype analysis (Supplementary Fig. 7a). Regardless of Fe availability, there were no differences in plant growth or nodule development, as well as in the response of SNF activity to H−N conditions (Supplementary Fig. 8a−g), except that the expression of *GmNRAMP2a* in the nodule of *nramp2b* mutant was significantly increased (Supplementary Fig. 8h). We therefore generated two double-knockout lines named *nramp2ab-1* and *nramp2ab-2* by CRISPR-Cas9 (Supplementary Fig. 7b, c). Phenotypic analysis showed that there was no difference in plant growth between WT and two double-knockout lines under non-symbiotic conditions (non-inoculation with H−N supply). However, after inoculated with rhizobium, the growth of two double-knockout lines was well below that of WT (Fig. 5a, b). Meanwhile, compared to WT plants, the double-knockout lines exhibited a significant decrease of 41% in single nodule weight (Fig. 5c, d), 48% in N export rate (Fig. 5e), and 46% in SNF activity (Supplementary Fig. 9a). Furthermore, the exogenous addition of high-Fe fully restored the nodule weight and SNF activity in double-knockout lines (Supplementary Fig. 9b, c), suggesting that the phenotypic defects in mutants are due to Fe limitation. Notably, *GmNRAMP2a&2b* knockout did not alter rhizobia invasion, nodule primordium initiation or nodule number per plant (Supplementary Fig. 9d−h). These results reveal that GmNRAMP2a&2b may not participate in the early-stage processes of rhizobia infection or nodule organogenesis, but rather affecting the later-stage processes of nodule development and SNF.

We next determined Fe status in nodules, and found that in the *nramp2ab* mutants, Fe intensity in fixation zone, as well as the Fe concentration in symbiosome were significantly decreased compared to the WT (Fig. 5f, g, Supplementary Fig. 9i). Subsequently, we isolated the intact uninfected and infected cell protoplasts through cell wall digestion and microcapillary separation (Supplementary Fig. 9j). Our results showed that Fe in the uninfected cell of *nramp2ab* mutants increased by 112%, while it decreased by 25% in the infected cell (Supplementary Fig. 9k), suggesting that GmNRAMP2a&2b are helpful for Fe delivery from uninfected cell to infected cell. To investigate whether GmNRAMP2a&2b affect the delivery of other trace elements in nodules, LA-ICP-TOF technology was used to observe the accumulation of trace elements in nodules. We found that the differences in Cu, Mn and Zn accumulation between WT and mutant nodules were not significant as compared to Fe (Supplementary Fig. 9l). Phenotypic analysis of WT and mutants under various micronutrient deficiency conditions revealed that the mutants exhibited unaltered biomass of nodule and seedling under Cu, Mn, or Zn-deficient conditions (Supplementary Fig. 9m, n). These findings indicate that GmNRAMP2a&2b play a more significant role in delivering Fe to nodules compared to other trace elements.

We used ferritin as a marker for intracellular Fe levels[10] and found decreased ferritin in *nramp2ab* mutants through transcriptomic and western blot assays (Supplementary Figs. 10, 11a; Supplementary Data 2), suggesting that defective Fe transport from vacuoles reduces cytoplasmic Fe. Conversely, *vtl1* mutants[21] showed increased ferritin (Supplementary Figs. 10, 11a; Supplementary Data 2), suggesting disrupted Fe transport to symbiosomes and resultant cytoplasmic Fe accumulation. In parallel, both *nramp2ab* and *vtl1* mutants showed reduced nitrogenase NifH and leghemoglobins (Lbs) levels

(Supplementary Figs. 10, 11a, Supplementary Data 2), indicating that disruption of Fe homeostasis (whether a deficiency or an excess) impair nodule N fixation.

We also examined how mutations affected the expression of gene families associated with Fe transport (*NRAMP*, *VIT*, *YSL*, *ZIP*, and *OPT*). The *nramp2ab* mutation led to decreased expression of three *NRAMP* genes, two *VIT* genes, one *YSL* gene, and two *ZIP* genes. In contrast, the *vtl1* mutation resulted in the reduced expression of two *NRAMP* genes, two *VIT* genes, and five *ZIP* genes, while also increasing the expression of two *NRAMP* genes, two *VIT* genes, and three *YSL* genes (including *GmYSL7*[28,29]), one *ZIP* gene, and one *OPT* gene (Supplementary Fig. 10). Overall, the effects of the *vtl1* mutation on the expression of Fe transporters are more pronounced than the *nramp2ab* mutation in nodules. To further validate these result, we expressed *GmNRAMP2b-GFP* in *vtl1* mutants, and found that GmNRAMP2b-GFP was undetectable in *vtl1* mutants (Supplementary Fig. 11b). These results suggest that Fe homeostasis in nodules relies on the complex interplay and coordination of multiple transporters.

### GmNRAMP2a&2b are involved in the inhibition of SNF by inorganic N

Due to the inhibitory effect of H−N on *GmNRAMP2a&2b* expression, we hypothesized that H−N might impact nodule Fe homeostasis through the regulation of GmNRAMP2a&2b. To test this hypothesis, we investigated the phenotypes of *nramp2ab* mutants under H−N conditions. We observed 38% and 35% inhibitions by H−N in N export rate in the WT and the mutants, respectively. Meanwhile, the WT showed a 33% reduction in Fe accumulation in H−N, whereas the mutants displayed only 16% reduction (Fig. 5e−g). H−N led to evident Fe bodies in uninfected cells of the WT, along with the appearance of lytic vacuoles in infected cells (Fig. 5h−j). In contrast, these Fe bodies were widespread in the mutants regardless of N levels, and H−N resulted in an increase in the number and size of lytic vacuoles compared to the WT (Fig. 5h-j). To further verify this hypothesis, we constructed overexpression lines and found that in both *GmNRAMP2a* and *GmNRAMP2a&2b* overexpression lines, although the plant seedlings and nodule phenotypes showed no significant changes under symbiotic conditions (Supplementary Fig. 12), H−N supply to the nodules led to significantly higher N export rate and Fe accumulation in fixation zone compared to the WT (Fig. 5k-l), while Fe bodies in uninfected cells and lytic vacuoles in infected cells was significantly lower than in the WT (Fig. 5m, n). Taken together, these results indicate that the inhibitory effect of H−N on SNF depends on GmNRAMP2a&2b.

### GmNIGT1a&1b negatively regulate N-responsive expression of GmNRAMP2a&2b

To investigate *cis*-acting elements responsive to N in the *GmNRAMP2b* promoter, we first constructed five vectors expressing *GUS* driven by different *GmNRAMP2b* promoter segments (P1-P5, Supplementary Fig. 13a). The transgenic nodules carrying each vector showed much lower expression of *GmNRAMP2b* after H−N supply for 1 d (Supplementary Fig. 13b), suggesting that −500 bp promoter region is enough for its N-responsive expression. Based on the known N-responsive *cis*-acting elements[19,30,31], we found three NIGT and one NIN-like *cis*-acting elements (named NIE and NRE respectively) in both −500 bp promoters of *GmNRAMP2a&2b* (Supplementary Fig. 13c, d). Considering the great importance of NIN/NLPs in root nodules, we first investigated their regulatory role on *GmNRAMP2a&2b* expression. Soybean has four NINs functioning redundantly in nodulation, and ten NLPs with unknown functions[32]. Since NINs and NLPs in soybean exist in the manner of paralogous gene pairs (Supplementary Fig. 14a), we selected one of each pair and overexpressed them in nodules for further investigation. Among these transgenic events, only overexpressing *GmNIN1b* reduced the expression of *GmNRAMP2a&2b* as well as single nodule weight (Supplementary Fig. 14b−d). However, GmNIN1b could

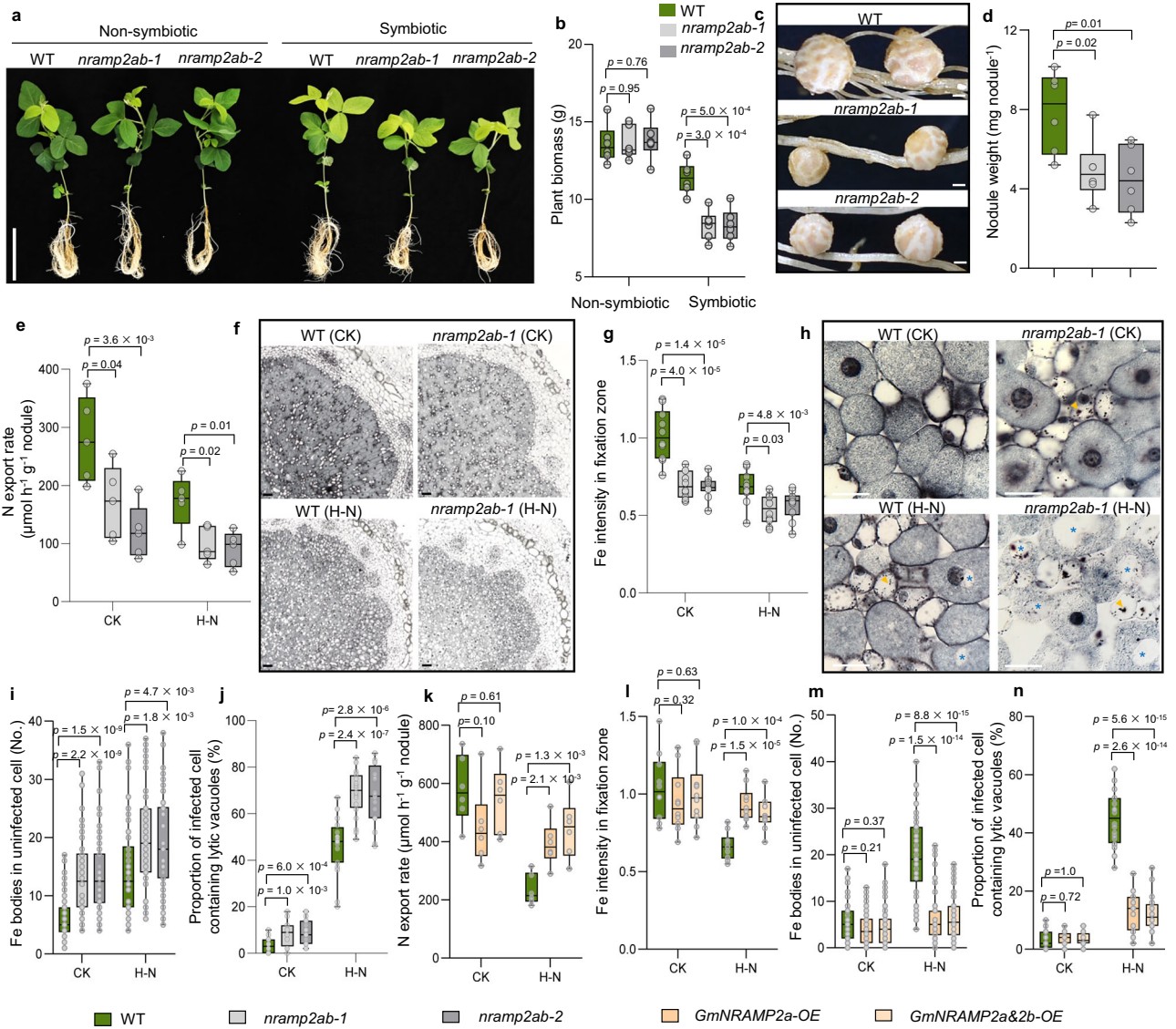

**Fig. 5 | GmNRAMP2a&2b are responsible for the Fe transfer for SNF, as well as the inhibition of SNF by inorganic N. a–d** Comparison of seedling (**a, b**) and nodule growth (**c, d**) in WT and *nramp2ab* mutants under non-symbiotic or symbiotic conditions. 4-day-old seedlings were grown under non-symbiotic (non-inoculation with high-N (H−N) supply) or symbiotic (inoculation with low-N (CK) supply) conditions for 22 d. **e–j** Comparison of ureide export rate (**e**), nodule Fe intensity (**f, g**), Fe bodies in uninfected cell (**h, i**), lytic vacuoles in infected cell (**j**) in WT and *nramp2ab* mutants. Nodules at 22 dpi were treated with or without H−N (CK) for 2 d. Nodule samples were sectioned and stained with Perls solution, followed by DAB intensification. Yellow arrows and blue asterisks in (**h**) indicate Fe bodies and lytic vacuoles, respectively. **k–n** Comparison of ureide export rate (**k**), nodule Fe intensity (**l**), Fe bodies in uninfected cell (**m**), lytic vacuoles in infected cell (**n**) in overexpression lines and their segregated WT. Nodules at 22 dpi were treated with or without H−N (CK) for 2 d. Nodule samples were sectioned and stained with Perls solution, followed by DAB intensification. The boxes in (**b, d, e, g, i–n**) indicate the first and third quartiles, and the whiskers indicate the minimum and maximum values. The lines within the boxes indicate the median values. *n* = 6 (**b, d, k**), 5 (**e**), 10 (**g, l**), 50 (**i, m**), or 20 (**j, n**) biologically independent replicates. Scale bars = 15 cm (**a**), 1 mm (**c**), 50 μm (**f, h**). Values were calculated using two-sided t-test. The phenotypic statistics for the overexpression lines were based on two independent lines, and for each treatment, at least three plants per line were evaluated. Source data are provided as a Source Data file.

bind to neither promoter of *GmNRAMP2a* nor *2b* by yeast one-hybrid assay (Supplementary Fig. 14e). Both GmNIN1b and AtNLP1[33] could bind to 4×NRE_AtNIR1 but not to 4×NRE_GmNRAMP2a/2b, suggesting that the predicted NRE_GmNRAMP2a/2b is not the true NIN/NLP binding site, and the reduced expression of *GmNRAMP2a&2b* might be an indirect effect due to the inhibited nodulation by *GmNIN1b* overexpression[30,32].

We next investigated the regulatory role of NIGT proteins on *GmNRAMP2a&2b* expression. There are four NIGTs (two paralogous gene pairs) in soybean with highest sequence homology to AtNIGTs in Arabidopsis (Supplementary Fig. 15a), but only one pair of them (named *GmNIGT1a&1b*) are dominantly expressed in nodules (Fig. 6a). Transcriptomic data revealed that similar to *GmNRAMP2a&2b*, both *GmNIGT1a&1b* were specifically expressed in nodules, and exhibited

much higher expression levels in fixation zone (Fig. 6b). Furthermore, both *GmNIGT1a&1b* were only up-regulated by H−N and had no response to Fe depletion (Fig. 6c). Their expression significantly increased in response to H−N from 7 to 50 dpi (Supplementary Fig. 2c, d), suggesting the role of GmNIGT1a&1b in N perception is active throughout early to late nodule development. This remarkable similarity in gene expression pattern indicates a potential complementary relationship of *GmNIGT1a* and *1b* in their biological functions.

We therefore selected *GmNIGT1a&1b* for yeast one-hybrid assay, and found that both of them fused with transcriptional activation domain (AD) of GAL4 can bind to the promoters of *GmNRAMP2a&2b*, as well as the NIGT cis-enriched sequence (6 × NIE, Fig. 6d). To verify

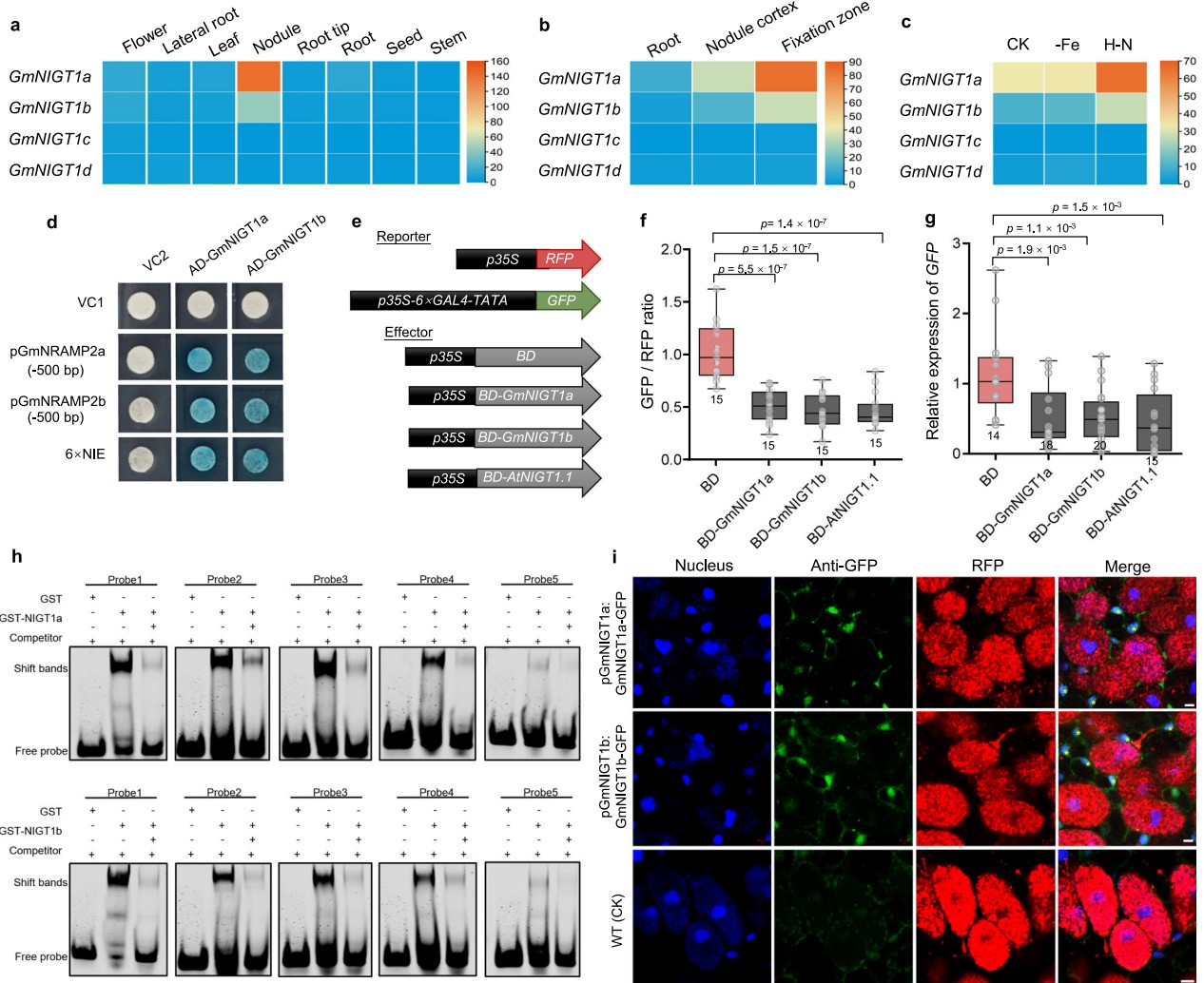

**Fig. 6 | The inhibitory effect of GmNIGT1a&1b on *GmNRAMP2a&2b*. a–c** Gene expression pattern of *GmNIGT1s*, including (**a**) Transcriptional abundance in tissues/organs. **b** Nodule-specific expression. **c** The expression response to Fe depletion (-Fe) or high-N (H–N) supply. Data in (**a**) are from Phytozome and Data in (**b**, **c**) are from RNA-seq described in Fig. 2. **d** Yeast one-hybrid assay. The effectors combined with the reporters were introduced into yeast strain EGY48 and cultured on SD medium (-Trp-Ura) containing X-gal at 30 °C. **e–g** Transcriptional repression effects of GmNIGT1a&1b in tobacco (**f**) and soybean hairy-root (**g**) after co-transformed with the effectors and the reporters in (**e**). The fluorescence signals were observed by microscope and GFP / RFP ratio were determined (**f**). Relative expression levels of *GFP* were determined by real-time RT-PCR (**g**). *EF-1α* was used as an internal standard. **h** EMSA showing GmNIGT1a&1b binding to the region of the *cis*-acting elements in the *GmNRAMP2a&2b* promoters. Plus (+) and minus (-) denote presence or absence of the probe or protein in each sample. The EMSA assay were performed with purified GmNIGT1a&1b and Cy5-labelled probes of

*GmNRAMP2a&2b* promoters. Unlabeled probes were used as competitors. GST alone was used as a negative control (lane 1). Probe positions are shown in Supplementary Fig. 13c. **i** Localization of GmNIGT1a&1b in nodule fixation zone. Nodules at 17 dpi from wild-type (WT, negative control) or transgenic hairy roots carrying *pGmNIGT1a/1b: GmNIGT1a/1b-GFP* were used for immunostaining. Blue shows signals from nucleus; Red shows RFP-tagged rhizobia; Green shows anti-GFP signals. The asterisks in merged images indicate blue and green overlapping parts. Five independent transgenic lines were investigated and consistent results were obtained, with one representative image presented. Scale bars, 5 μm. The boxes in (**f**, **g**) indicate the first and third quartiles, and the whiskers indicate the minimum and maximum values. The lines within the boxes indicate the median values. The number of biologically independent replicates (n) is indicated at the bottom of each box in the figures. The values (**f**, **g**) show significant differences compared with BD. Values are the mean ± SD (two-tailed *t*-test). Source data are provided as a Source Data file.

whether GmNIGT1a&1b have the same repression effects on gene transcription as AtNIGTs, we employed transcriptional repressor activity assays in both tobacco leaves and soybean nodules. The effector (GAL4 binding domain (BD)-fusion protein) is able to bind with the GAL4 cis-enriched sequence (6×GAL4) upstream of the reporter gene (*GFP*), and thereby affects *GFP* expression if the effector has a transcriptional activation/repression activity. The results revealed that in both systems (tobacco leaves and soybean nodules), expression of *GmNIGT1a*, *GmNIGT1b* or *AtNIGT1.1* individually led to a significant reduction in the expression of reporter gene (Fig. 6e–g). This indicates that GmNIGT1a&1b may function as transcriptional repressors in soybean.

On the other hand, EMSA result showed that both GmNIGT1a and 1b could directly bind to all five of putative *cis*-acting elements in vitro, although their binding affinity to probe 5 was markedly weaker (Fig. 6h, Supplementary Fig. 13c). In situ immunostaining results showed that both GmNIGT1a and 1b were localized in all nodule cells except for the infected cell, with high-intensity signals at nuclei and weak signals at cytoplasm (Fig. 6i, Supplementary Fig. 15b–d).

## GmNIGT1a&1b regulate Fe homeostasis in nodules

To clarify the role of GmNIGT1a&1b in nodule Fe homeostasis and SNF, we constructed the double knockout line (*nigt1ab-cr*) by CRISPR-Cas9 (Supplementary Fig. 7d), as well as the knockdown (*nigt1ab-RNAi*) and

overexpression lines (*GmNIGT1a-OE*). There was no change in the growth of seedlings and nodules in the *nigt1ab-RNAi* and *nigt1ab-cr* lines, while *GmNIGT1a-OE* lines displayed notable decreases in seedling and nodule growth (Fig. 7a–d). Knockdown or knockout of *GmNIG-T1a&1b* alleviated the inhibition of *GmNRAMP2a&2b* expression by H–N, while overexpressing *GmNIGT1a* resulted in a low-level expression of *GmNRAMP2a&2b* regardless of N availability (Fig. 7e–f). When inorganic N was supplied to the nodules, *nigt1ab-RNAi* and *nigt1ab-cr* lines exhibited higher N export rates and Fe accumulation in nodules than WT (Fig. 7g-j). In contrast, *GmNIGT1a-OE* lines displayed notable reductions in N export rates and Fe accumulation in fixation zone (Fig. 7g–j). In parallel, the number of Fe bodies in uninfected cells and lytic vacuoles in infected cells was significantly lower in *nigt1ab-RNAi* and *nigt1ab-cr* lines but higher in *GmNIGT1a-OE* lines (Fig. 7k–m).

*p35S*-driven overexpression is not limited to nodules and can potentially lead to indirect effects on nodule development. To achieve in situ overexpression, we utilized the *GmNIGT1a* promoter to drive the expression of *GmNRAMP2b* and the *GmNRAMP2b* promoter to drive *GmNIGT1a* expression in soybean hairy roots. As expected, nodules with *pGmNIGT1a:GmNRAMP2b-GFP* exhibited a stronger GFP signal under H–N conditions, and the Fe accumulation and distribution in these nodules followed a trend similar to nodules with *p35S:GmNRAMP2b*. Conversely, nodules containing *pGmNRAMP2b:Gm-NIGT1a-GFP* showed a stronger GFP signal in control conditions, and the Fe accumulation and distribution mirrored the pattern seen in nodules with *p35S:GmNIGT1a* (Supplementary Fig. 16). Taken together, these observations indicate a direct and specific regulatory role of NIGT1-NRAMP2 module in Fe homeostasis and N-inhibited SNF within nodules.

To explore more downstream targets of GmNIGT1a&1b, we conducted a comparative transcriptomic assay, identifying 47 genes potentially regulated by GmNIGT1a&1b (Supplementary Data 3). These genes are up-regulated in *nigt1ab* double knockout line, but down-regulated in *GmNIGT1a-OE* line and under H–N conditions (Supplementary Fig. 17a). Notably, our findings reveal that, in addition to *GmNRAMP2a&2b*, putative Fe homeostasis-related genes such as *IRON MAN* (*IMA*s) and *BRUTUS*s are also under the regulation of GmNIG-T1a&1b (Supplementary Fig. 17b). Whether they are involved in Fe homeostasis requires further study.

## Discussion

Recent reports have revealed that VTL transporters facilitate Fe transport into symbiosomes for SNF in legume nodules[21–23]. Although previously GmNRAMP2b, also termed GmDMT1, was presumed to act as an Fe transporter at the SM of soybean nodules[20], this has been brought into question by researchers which suggest it is unlikely to be involved in exporting Fe out of cells[34,35]. Given that the physiological role of GmNRAMP2b in SNF remains unconfirmed, its contribution to Fe homeostasis within nodules was not yet understood. Furthermore, GmNRAMP2b and GmVTL1a showed different transcriptional response to N and Fe availability ([21]; Fig. 2g–i), suggesting their distinct physiological functions. In this study, we cannot verify the previously reported results that GmNRAMP2b locates at the SM of infected cell[20]. Instead, we demonstrate that GmNRAMP2b and its paralog GmNRAMP2a works primarily in uninfected tissues (Fig. 3a–f, Supplementary Fig. 3a–f), and whether expressed in yeast, tobacco protoplasts or soybean nodules, GmNRAMP2a&2b were localized at the tonoplast (Fig. 3g–j, Supplementary Fig. 4a). Besides, analyses of two single-cell databases (zhailab.bio.sustech.edu.cn/single cell soybean; soybeancellatlas.org) consistently indicate that GmNRAMP2a&2b and GmNIGT1a&1b are predominantly found in the uninfected cells of soybean nodules. Moreover, upon investigating the published soybean SM proteomics data[36], we found no detection of GmNRAMP2a, 2b, GmNIGT1a, or 1b proteins. Nevertheless, *GmNRAMP2b* was still expressed at a low level in infected cells (Fig. 3d–f), suggesting its potential minor role in these cells.

GmNRAMP2b was considered as a Fe influx transporter due to its ability of rescuing the growth of the *fet3fet4* mutant defective in ferrous Fe uptake[20]. However, GmNRAMP2b was found to be non-PM localized (Supplementary Fig. 4a), which led us to reason that complementation of *fet3fet4* strain by GmNRAMP2b was probably caused by an indirect effect. The yeast system does have some limitations when used to express exogenous proteins, such as the mislocalization of plant chloroplast membrane proteins to the yeast cell membrane[37]. However, it is a commonly used system for gene functional complementation tests[38]. Therefore, by ectopically expressing *GmNRAM-P2a&2b* on the yeast plasma membrane, we were able to successfully complement the *fet3fet4* mutant (Fig. 4c–e), suggesting that GmNRAMP2a&2b act as transporters for ferrous Fe influx, and in planta, they facilitate the transport of Fe from the vacuole to the cytoplasm.

In soybean nodules, uninfected cells play an important role in supporting nitrogen fixation. However, little attention has been paid to these uninfected cells, and it was previously thought that they were only involved in C and N metabolism[39,40]. In this study, we observed that compared to WT plants, *nramp2ab* mutants accumulated less Fe in infected cells and symbiosomes, but more Fe in uninfected cells (Fig. 5f–i, Supplementary Fig. 9i–k). This suggests that GmNRAM-P2a&2b can mobilize Fe from the vacuoles of uninfected cells, which can then be transported to the apoplast and subsequently enter infected cells via other Fe transporters ([41], Fig. 8). Alternatively, the mobilized Fe could be transported via plasmodesmata through the symplastic pathway, as suggested by previous research ([42], Fig. 8). Infected cells in soybean nodules are mostly occupied by rhizobia, and have no vacuoles for nutrient storage, while uninfected cells have large central vacuoles and provide a large surface area for interaction with the infected tissue[43]. Therefore, GmNRAMP2a&2b in uninfected cells can ensure timely and dynamic release of Fe from vacuoles to infected cells for SNF.

While *nramp2ab* mutants exhibited a reduced nodule size and nitrogenase activity, their phenotype is not as severe as for *vtl1* mutants, which show little activity for SNF[21]. We reason that in addition to GmNRAMP2a&2b-involved Fe transport, other Fe transport pathways and relevant transporters are also important for Fe delivery to infected cells (Supplementary Fig. 10). For example, a NRAMP homolog from *Medicago truncatula* is localized at the PM of infected cells, responsible for transport of Fe from the apoplastic into infected cells[41]. The equivalent transporter in soybean may collaborate with NRAMP2 to ensure Fe homeostasis in nodules. In general, VTL1 and NRAMP2 proteins play different roles in soybean nodules. VTL1 is an indispensable protein for maintaining the basic N fixation of infected cells, while NRAMP2 predominantly functions as a regulator in uninfected cells, with its role becoming more pronounced when the nodule has a higher demand for Fe.

Organisms have developed genetic robustness to maintain normal development in response to harmful mutations. In addition to gene redundancy, genetic compensation response is recently suggested as another important mechanism for genetic robustness, where one or more paralogs are upregulated to substitute for the compromised activity of another[44,45]. In this study, we found that the paralogs *GmNIGT1a* and *1b* exhibited highly similar expression patterns (Fig. 6a–c), suggesting they may have complementary functions. In contrast, although GmNRAMP2a&2b shared the same Fe transport activities and subcellular localizations (Figs. 3, 4), they differed in expression patterns. Compared to the dominant expression of *GmNRAMP2b*, *GmNRAMP2a* was low-level expressed and upregulated only under conditions of Fe-depletion or sole *GmNRAMP2b* knockout (Fig. 2f, g, Supplementary Fig. 8 h). This suggests that GmNRAM-P2a&2b possess an asymmetrically redundant role in Fe transport. Instead, GmNRAMP2a has evolved to provide an active dosage compensation when large amounts of Fe are required in nodules. As

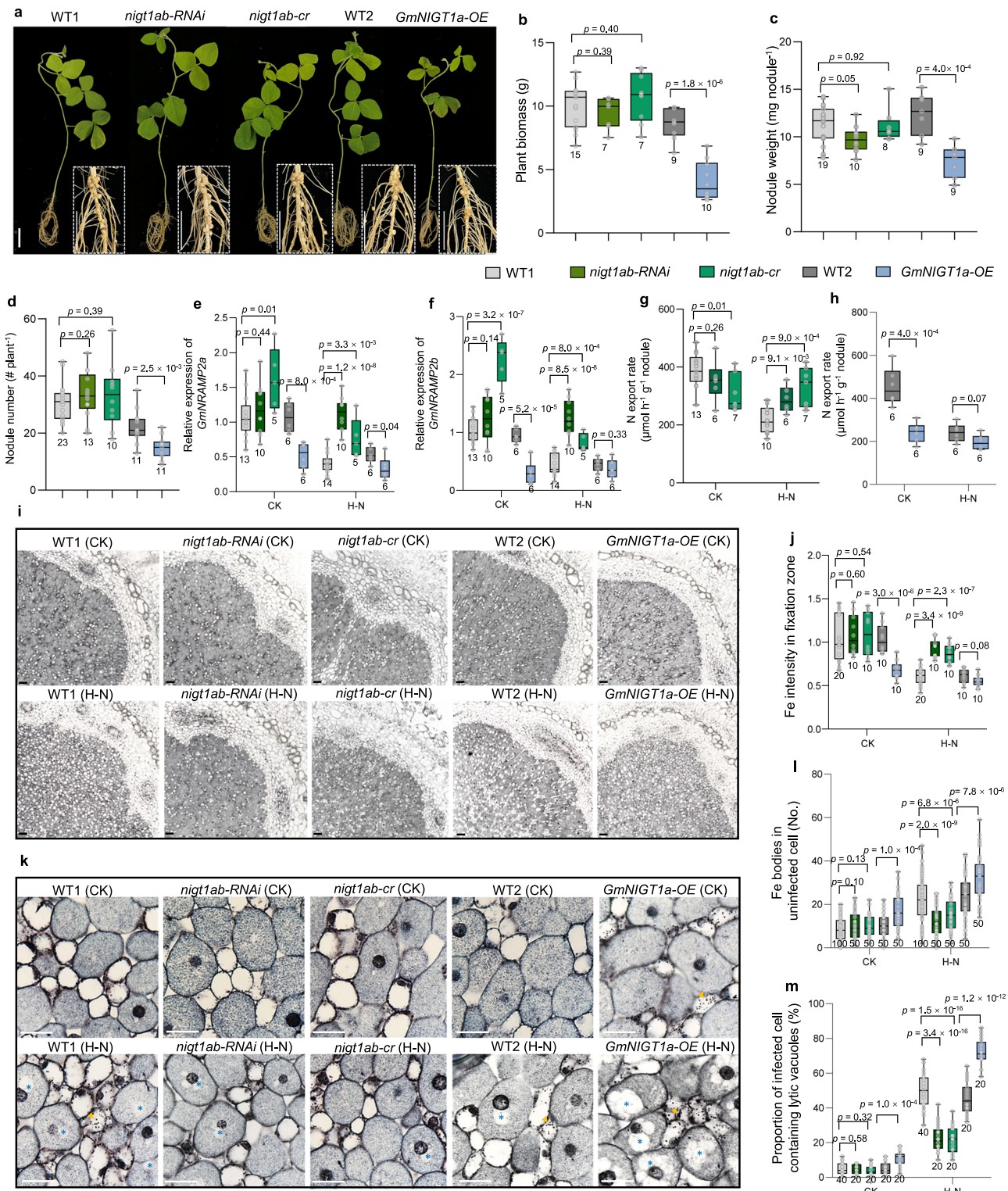

**Fig. 7 | GmNIGT1a&1b regulate Fe homeostasis in nodules. a–d** Seedling and nodule growth in knockdown (*nigt1ab-RNAi*), knockout (*nigt1ab-cr*) and overexpression (*GmNIGT1a-OE*) lines. **a** Whole-plant growth. Scale bars, 2 cm. **b** Plant biomass. **c** Nodule weight. **d** Nodule number. 4-day-old seedlings were inoculated with rhizobia and grown in a low-N nutrient solution for 21 d. WT1, the wild-type control for *nigt1ab-RNAi* and *nigt1ab-cr*. WT2, the wild-type control for *GmNIGT1a-OE* lines. **e–f** Gene expression of *GmNRAMP2a* (**e**) and *2b* (**f**) in nodules of *nigt1ab-RNAi*, *nigt1ab-cr* and *GmNIGT1a-OE* lines. Nodules at 21 dpi were treated without (CK) or with high-N (H-N) solution for 1 d. Relative expression levels were determined by real-time RT-PCR. *EF-1α* was used as an internal standard. **g–m** Ureide export rate (**g**, **h**), nodule Fe intensity (**i**, **j**), Fe bodies in uninfected cell (**k**, **l**), lytic vacuoles in infected cell (**m**) of *nigt1ab-RNAi*, *nigt1ab-cr* and *GmNIGT1a-OE* lines.

Nodules at 21 dpi were treated without (CK) or with high-N (H-N) solution for 2 d. Nodule samples were sectioned and stained with Perls solution, followed by DAB intensification. Yellow arrows and blue asterisks in (**k**) indicate Fe bodies and lytic vacuoles respectively. Scale bars = 1 mm (**i**), 50 μm (**k**). The boxes in (**b–h**, **j**, **l**, **m**) indicate the first and third quartiles, and the whiskers indicate the minimum and maximum values. The lines within the boxes indicate the median values. The number of biologically independent replicates (*n*) is indicated at the bottom of each box in the figures. The *P* values show significant differences compared with WT. All *P* values were calculated using two-sided t-test. The phenotypic statistics for the overexpression and RNAi materials were based on two independent lines, and for each treatment, at least three plants per line were evaluated. Source data are provided as a Source Data file.

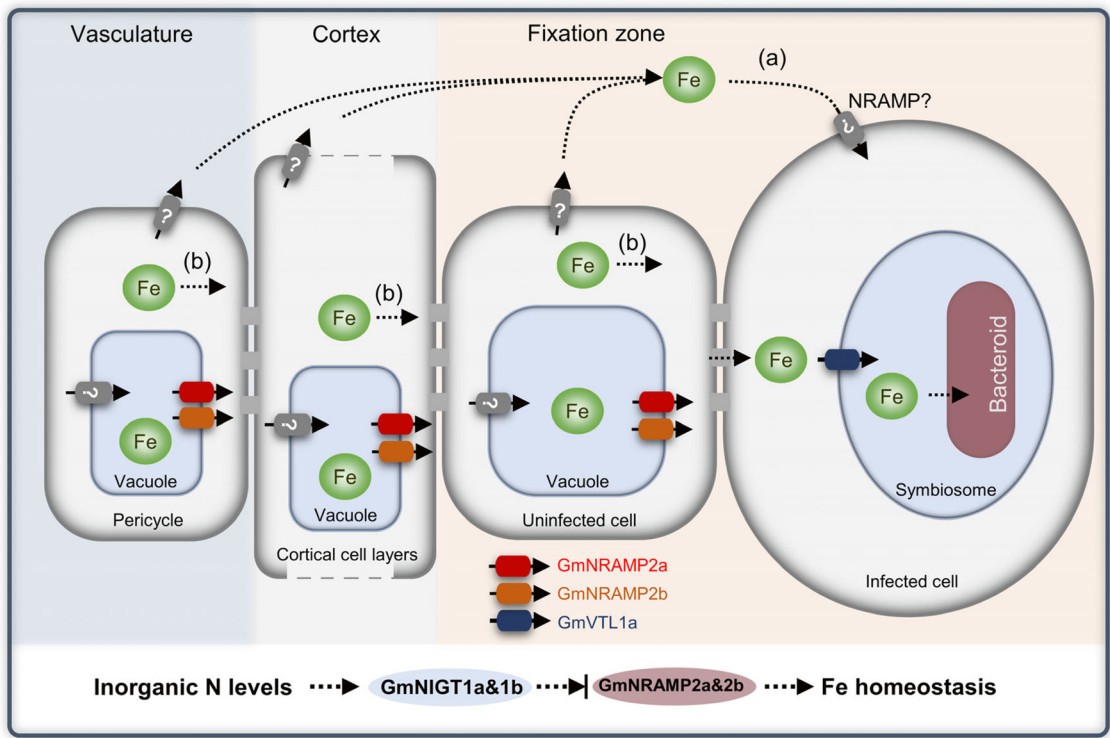

**Fig. 8 | Proposed model of NIGT1-NRAMP2 regulatory module in soybean nodules.** Fe is unloaded from nodule vasculature to infected cells by apoplastic (**a**) and symplastic (**b**) approaches. GmNRAMP2a&2b are both Fe influx transporters localized at the tonoplast of uninfected tissues, and they facilitate Fe transfer to infected cells. Under inorganic N-sufficient conditions, the expression of *GmNRAMP2a&2b* is suppressed by a pair of N-signal regulators GmNIGT1a&1b in uninfected tissues, and thus blocks Fe transfer for SNF. GmVTL1a facilitates Fe transport into symbiosomes for SNF[21].

soybean is an ancient tetraploid with ~75% of current genes present in multiple copies, and *GmNRAMP2a&2b* are paralogs resulting from genome duplication events[24], our study offers functional evidence supporting the notion that genome duplication enhances soybean's environmental adaptability.

It was found that both *GmNRAMP2a&2b* were regulated by N and Fe availability in nodules (Fig. 2). For Fe-regulation, it seems reasonable because these two genes encode Fe transporters, and similar regulation of *NRAMP* genes by Fe availability as well as their regulatory networks have been widely reported in other species[46]. For N-regulation, it seems reasonable particularly for nodule organs as Fe is indispensable for SNF, and SNF is tightly controlled by external inorganic N levels. However, N-modulated Fe homeostasis is little understood. Intriguingly, our study revealed a NIGT-NRAMP regulatory module in nodules. NIGT family proteins are a group of G2-like GARP-type transcription factors that were previously shown to suppress expression of a series of genes related to nitrate transport and assimilation, and are thereby recognized as N-satiation-signal transducers to prevent excessive N accumulation and energy consumption[47]. The existence of a NIGT-NRAMP regulatory module suggests that a novel function obtained by soybean NIGTs in the symbiotic system, is to regulate Fe transport to achieve dynamic Fe supply for SNF.

How NIGT perceive and transmit N signals within nodules remains unknown. However, it is interesting to note that most NIGT1s are identified as direct targets of NAC transcription factors[48]. This suggests that NACs could at least partially mediate the transmission of N signals through the NIGT1 signaling pathways. Meanwhile, the genes downstream of NIGT1 include not only *NRAMP2* but also *IMAs* and *BRUTUSs* (Supplementary Fig. 17b). IMA peptides positively regulate Fe homeostasis in plants by interacting with the E3 ubiquitin ligase BRUTUS, which is required for the degradation of transcription factors involved in the Fe deficiency response[49,50]. In root nodules of *Lotus japonicus*,

IMA peptides have recently been reported to regulate nitrogen fixation[51], which highlights the essential role of IMA-mediated Fe provision in regulating N-related physiological processes. Whether IMA peptides regulate NRAMP2 or act independently to maintain Fe balance in root nodules requires further study.

Nodule formation and nitrogen fixation require a lot of energy, and legumes have developed strategies to adjust nodule numbers and SNF levels in response to changes in N levels in the environment[18]. As Fe is vital for both the host and rhizobia, legumes may save resources by preventing Fe allocation to SNF via NIGT perception of N signals when sufficient N is available to plants. Therefore, in agricultural fertilization management, it is essential to emphasize the supplementation of the trace element - Fe, to mitigate the inhibition of nodule SNF by inorganic nitrogen fertilizers.

## Methods
### Plant material and growth conditions
The stable gene knockout mutants were obtained using CRISPR-Cas9 technology in the soybean (*Glycine max*) genotype Williams 82[52]. The guide RNA sequence for each mutant is shown in Supplementary Data 4. Transgenic seedlings were then generated through *Agrobacterium tumefaciens* (EHA105)-mediated transformation[53]. The predicted editing sites in T1 seedlings were sequenced and those with frameshift mutations were selected. T2 homozygous seeds were collected for phenotypic analysis.

To generate stable *GmNIGT1a&1b* knockdown material (*nigt1ab-RNAi*), a 273-bp (position 622-894 starting from ATG) conserved region of *GmNIGT1a&1b* with 96% nucleotide identity, was amplified and inserted into the *Asc*I and *Swa*I sites of pFGC5941 in the sense orientation. This construct was then inserted into the *Xba*I and *BamH*I sites in the anti-sense orientation. 35S promoter was used for the RNAi construction.

To construct the stable overexpression lines, the ORFs of *GmNRAMP2a*, *GmNRAMP2b*, and *GmNIGT1a* were amplified and individually inserted into the *Asc*I and *Xba*I sites of pFGC5941-*p35S* construct. 35S promoter was used for the overexpression line construction. Transgenic seedlings were then generated through *Agrobacterium tumefaciens* (EHA105)-mediated transformation. *GmNRAMP2a&2b* double overexpression lines were obtained by co-transformation.

Soybean seeds were surface-sterilized by exposure to chlorine gas overnight prior to germinating in sterilized vermiculite. After 4 d, seedlings were inoculated with *Bradyrhizobium* strain BXYD3 or RFP-expressing strain[21], and cultured with a low-N nutrient solution in vermiculite. Seedlings were then transplanted and cultivated in a low-N nutrient solution before various Fe or N treatments. Low-N solution was prepared with 1/10 of the N in the base nutrient solution (5.3 mM)[54]. High-N solution was supplemented with $NH_4NO_3$ to achieve a total N concentration of 20 mM. EDTA-Fe (10 μM) was used for plant culture if not otherwise specified.

To generate transgenic soybean composite plants, the hypocotyl injection method for hairy root transformation was utilized according to ref. 21. The transformed hairy roots from 25-d-old seedlings were inoculated with *Bradyrhizobium* strain and cultured in a low-N nutrient solution before Fe or N treatment.

All seedlings were grown in a growth chamber with a 13 h/26 °C day and 11 h/24 °C night regime, with daytime light provided by light-emitting diode at an intensity of 400 μmol photons $m^{-2}s^{-1}$, and relative humidity maintained at 65%. Nutrient solutions were renewed every 2 days and pH was adjusted to 5.8. Roots were continuously aerated through an air pump.

## Perls/DAB staining
Nodule samples were embedded in resin according to the method of ref. 55. Briefly, Nodules were incubated overnight in fixation solution containing 50% (v/v) ethanol, 5% (v/v) glacial acetic acid, and 10% (v/v) formaldehyde solution and vacuum infiltrated for 30 minutes. The fixed nodules were dehydrated in a series of 50%, 60%, 70%, 80%, and 90% ethanol solutions, and then overnight dehydrated in 100% ethanol. Samples were embedded in Technovit 7100 resin (Kulzer) according to the kit instructions, and thin sections (7 μm) were prepared. These sections were vacuum infiltrated for 15 min each with equal volumes of 4% (v/v) HCl and 4% (w/v) K-ferrocyanide (Perls stain solution), and incubated for 30 min at room temperature. For DAB intensification, fixed sections were washed with distilled water and incubated in a methanol solution containing 10 mM $NaN_3$ and 0.3% (v/v) $H_2O_2$ for 1 h. After washing with PBS, sections were then incubated in an intensification solution containing 0.025% (w/v) DAB, 0.005% (v/v) $H_2O_2$, 0.005% (w/v) $CoCl_2$ and 0.1 M PBS (pH 7.4) for 30 min prior to being washed with distilled water to stop the reaction. These sections were photographed using an optical microscope (Nikon Ni-U, Japan).

## Isolation of intact symbiosomes
Intact symbiosomes were isolated according to ref. 21. Briefly, fresh nodules were ground gently in an ice-cold homogenizing buffer. Samples were then filtered through 4 layers of miracloth (Millipore, USA), and slowly transferred onto the top of a 30/60% (v/v) Percoll gradient solution. After centrifuging at 4000 g for 15 min, symbiosomes were collected from the 60% Percoll fraction (including the 30/60% interface). Collected symbiosomes were rinsed three times with a wash buffer. The number of symbiosomes was counted using a hemocytometer under a light microscope (Primo Star, Carl Zeiss, Germany). Samples were then digested in concentrated nitric acid for measurement of Fe concentrations using ICP-MS (Agilent 7900, USA).

## N export rate and acetylene reduction assay
Basal regions of soybean shoots (2 cm above the roots) were excised with a razor, and then xylem sap was collected for 1 h, and the concentration of ureides was determined using colorimetric analysis of glyoxylate derivatives according to the ref. 54. N export rate of nodule was calculated as the total ureide content divided by the fresh weight of nodules.

Acetylene reduction activity in nodules was determined according to ref. 54. Briefly, nodules were isolated and kept in an air tight glass bottle, and then immediately exposed to acetylene gas for 2 h. After injecting 0.5 M NaOH to terminate the reaction, a 0.3 mL gas sample was extracted and injected into a gas chromatograph (GC-2014, SHIMADZU, Japan) for ethylene determination.

## Transcriptomic analysis
For the transcriptomic analysis depicted in Fig. 2a–c, the -Fe treatment involved transplanting seedlings at 10 dpi into a low-N and Fe-free solution for 7 days. The H−N treatment entailed initially transplanting seedlings at 10 dpi into a low-N solution and EDTA-Fe ( + Fe) for 6 days, followed by exposure to H−N solution for 1 day. The CK treatment involved transplanting seedlings at 10 dpi into a low-N and +Fe solution for 7 d. For the transcriptomic analysis depicted in Supplementary Fig. 10, the -Fe treatment involved transplanting seedlings at 14 dpi from WT, *nramp2ab*, and *vtl1* mutants into a low-N and Fe-free solution for 7 days. The H−N treatment involved transplanting seedlings at 20 dpi into a H−N solution for 1 d. Low-N and +Fe treatments were used as CK. For the transcriptomic analysis depicted in Supplementary Fig. 17, nodules grown in a low-N solution at 21 dpi from WT, *nigt1ab-cr* and *GmNIGT1a-OE* lines were used for RNA-seq analysis. The H−N treatment involved transplanting seedlings of WT at 20 dpi into a H−N solution for 1 d. Nodule samples were harvested and quickly frozen by liquid nitrogen for subsequent RNA sequencing analysis using an Illumina HiSeqTM 2500 platform (Novogene, China). Genes with fold change larger than 2 (or $log_2$ FC > 1) were selected.

## Quantitative gene expression analysis
To investigate the gene expression in nodules, nodules at 21dpi were separated into three parts for RNA extraction: nodule conjugated root segments with nodules removed, nodule cortex, and fixation zone[21]. For time-course analysis, nodules grown in a low-N solution were harvested for RNA extraction at 7, 14, 17, 21, 30, 40, and 50 dpi. To investigate the expression response to various nutrient stresses, nodules grown in a low-N solution at 10 dpi were transferred to a low-N and Fe-, Mg-, Mo-, Mn-, Zn- or S-free solution for 7 d, or nodules grown in a low-N solution at 16 dpi were treated with high-N for 1 d, and then were harvested for RNA extraction. To investigate the expression response to N and Fe interaction, nodules grown in a low-N solution at 21dpi were treated with H−N, -Fe or a combination of both for 1, 2, 3 or 4 d. To investigate the expression response to different N source, nodules grown in a low-N solution at 21dpi were treated with 10 mM ammonium, 10 mM nitrate or a combination of both for 1, 2, 3 or 4 d.

For real-time reverse transcription (RT)-PCR, total RNA was extracted using *TransZol* Up Plus RNA Kit (TransGen, China). 500 ng of RNA was used for complementary DNA (cDNA) synthesis using TransScript One-Step genomic DNA Removal and cDNA Synthesis Super Mix (TransGen, China). Gene expression levels were determined by real-time RT-PCR using TransStart Top Green qPCR SuperMix (TransGen, China). The housekeeping gene EF-1a was used as an internal control. Normalized relative expression was calculated by the ΔΔCt method. The primers used for RT-PCR are shown in Supplementary Data 4.

## Phylogenetic analysis
Protein sequences were obtained from Phytozome (phytozome-next. jgi.doe.gov/) and miyakogusa.jp (kazusa.or.jp/lotus) database. The alignment analysis of protein sequences was performed using MEGA7.

## Tissue and subcellular localization

To investigate tissue-specific expression of *GmNRAMP2a&b*, their respective 2.5 kb promoter sequences were amplified and cloned into the pFGC5941-*GFP* vector to create the *pGmNRAMP2a:GFP* and *pGmNRAMP2b:GFP* constructs. To determine subcellular localization of GmNRAMP2a&b proteins in nodules, the ORFs of *GmNRAMP2a&b* were individually amplified and inserted into the above constructed vectors to create the *pGmNRAMP2a:GmNRAMP2a-GFP* and *pGmNRAMP2b: GmNRAMP2b-GFP* constructs. To determine subcellular localization of GmNIGT1a&1b proteins in nodules, sequences including 2.5 kb upstream promoter and genomic gene sequence were amplified and cloned into the pFGC5941-*GFP* vector to create the *pGmNIGT1a: GmNIGT1a-GFP* and *pGmNIGT1b: GmNIGT1b-GFP* constructs. The primers are shown in Supplementary Data 4. These constructs were transformed into *Agrobacterium rhizogenes* strain K599 for hairy-root transformation. The transformed hairy roots from 25-d-old seedlings were inoculated with rhizobia and cultured in a low-N nutrient solution before Fe or N treatment. Nodules at 17dpi or 30 dpi were collected for the immunostaining. Immunostaining was performed according to the methods of Liu et al. [21]. A polyclonal anti-GFP (1:1000; Thermo Scientific, USA) was used for the primary antibody. Alexa Fluor 488 or 555 goat anti-rabbit IgG (1:2000; Thermo Scientific, USA) were used for the secondary antibody. Calcofluorwhite (1:2000; Sigma, USA) and DAPI (1:500; Solarbio, China) were used for cell wall and nucleus staining, respectively.

To investigate the subcellular localization of GmNRAMP2a&2b in tobacco (*Nicotiana tabacum*) protoplasts, the ORFs of both genes were amplified and then inserted into pFGC5941-*p3SS-GFP* to obtain *p35S:GmNRAMP2a-GFP* and *p35S:GmNRAMP2b-GFP*. FM4-64 FX (Thermo Scientific, USA) was used as a PM marker. The protoplasts used for transient expression analysis were extracted from tobacco grown in Fe-sufficient conditions and transformed by the polyethylene glycol (PEG) method[56]. To investigate the subcellular localization of GmNRAMP2a&b in yeast, *GmNRAMP2a-GFP* and *GmNRAMP2b-GFP* sequences were amplified and cloned into pYES2 vector (V82520, Invitrogen, USA) respectively. Subsequently, the PM signal peptide ENO2(169)[27] was amplified from yeast DNA and inserted in front of *GmNRAMP2a/2b*-GFP. The primers are shown in Supplementary Data 4. The reconstructed vectors were transformed into wild-type strain BY4741 using the S.c.easy Comp Transformation Kit (Invitrogen, USA). Fluorescence was observed with a confocal scanning microscope (LSM880, Carl Zeiss, Germany) after yeast growth with galactose.

## Immunoelectron microscopy

The nodules samples were fixed with 4% paraformaldehyde (PFA) in phosphate buffer (PB, 0.1 M, pH 7.0) for 30 min followed by agar embedding and oscillating slicing. The oscillating sections (120-150 μm) were rapidly frozen and fixed in a high pressure freezing apparatus (Wohlwend Compact 03, Wohlwend, Switzerland), and then transferred to 0.2% uranyl acetate in pure acetone at −90°C for subsequent freeze substitution in a freeze substitution instrument (EM AFS2, Leica, Germany). Then the frozen water in the samples was gradually replaced by acetone and resin Lowicryl HM20 (Electron Microscopy Sciences, USA) at −45°C. Embedding and UV polymerization were performed stepwise at −40°C. For Immunoelectron microscopy, the ultrathin sections were immunolabled with anti-GFP antibody (1:50, Abcam, UK) as primary antibody for 90 min, followed by treatment with goat anti-rabbit IgG conjugated with 15-nm-diameter gold particles as secondary antibody (1:100, Abcam, UK) for 60 min. Sections were then stained and observed using TEM (HT7800, Hitachi, Japan).

## Western blot analysis

For western blot of nitrogenase, ferritin and leghemoglobin, nodules from WT, *nramp2ab* and *vtl1* mutants were harvested and ground into powder in liquid N. Sample was loaded equally onto an SDS-PAGE gel, and then blotted to a polyvinylidene fluoride membrane (Immobilon-P, Millipore, USA). The membrane was probed with anti-NifH (1:2000; Agrisera, Sweden), anti-ferritin (1:2500; Agrisera, Sweden), or anti-leghemoglobin[57], anti-actin (1:5000; ABclonal, China) overnight, and followed with their corresponding horseradish peroxidase (HRP)-conjugated second antibodies (anti-chicken IgY (1:10,000; Thermo Scientific, USA) for NifH, anti-rabbit IgG (1:5000; Biosharp, China) for ferritin; anti-Goat IgG (1:1000; Solarbio, China) for leghemoglobin, anti-mouse IgG (1:5000; TransGen, China) for actin) for 1 h. For western blot of yeast marker proteins, anti-ALP (1:1000; Abcam, UK), anti-PGK (1:2000; Abcam, UK) and anti-porin (1:1000; Abcam, UK) were used as primary antibodies. Anti-mouse (for ALP and porin) and anti-rabbit (for PGK) IgG HRP-conjugated antibody (1:2000; TransGen, China) were used as second antibodies. The HRP signals were detected using the SuperSignal West Dura Trial Kit (Thermo Scientific, USA) with an Amersham Imager 600 (GE Healthcare Bio-Sciences AB, Sweden). For full scan blots, please see the Source Data file.

For western blot of nodules' membrane proteins, intact symbiosomes were isolated as described above. Subsequently, symbiosomes were separated into symbiosome membrane (SM), symbiosome space (SS) and bacteroids (B) according to ref. [58]. The supernatant from percoll gradient centrifugation were further fractionated using discontinuous sucrose gradients (20–60%) according to ref. [36]. Immunoblot analysis was performed using primary antibodies for GFP (1:1000; TransGen, China), V-type ATPase (1:2000; Agrisera Sweden), H⁺-ATPase (1:2000; Agrisera Sweden), Nodulin-26 (1:500; the synthetic peptide TKNTSETIQRSDSLV was used to immunize rabbits to obtain antibodies against Nodulin-26). Anti-mouse (for GFP) and anti-rabbit (for V-type ATPase, H⁺-ATPase, and Nodulin-26) IgG HRP-conjugated antibody (1:2000; TransGen, China) were used as second antibodies.

## Complementation test in yeast

The amplified ORFs of *GmNRAMP2a*, *GmNRAMP2b*, *GmVTL1a* or full-length cDNA of *GmNRAMP2b* were cloned into pYES2 vector, which was then transformed into BY4741 (WT) or *Δccc1* yeast strain[26] using the S.c.easy Comp Transformation Kit (Thermo Scientific, USA). The primers are shown in Supplementary Data 4. Yeast transformants were selected on synthetic defined medium without Ura (SD-Ura) containing 2% glucose. After liquid culture with glucose to exponential phase, yeast transformants were incubated with SD-Ura containing 2% galactose, 1% raffinose and 1 mM FeSO₄ for 0, 0.5, 1 or 2 h.

Vacuoles were isolated according to the methods of Li et al.[26] with the following modifications: 300 ml of yeast cells was collected by centrifugation at 3000 g for 3 min. The cells were resuspended in 10 ml of 0.1 M Tris-HCl (pH 9.3) and 10 mM dithiothreitol, and incubated for 10 min at 30 °C. The cells were washed once with spheroplast buffer (1.2 M sorbitol, 20 mM potassium phosphate, pH 7.4) and incubated with 500 U/ml lyticase (Solarbio, China) for 2 h at 30 °C. Spheroplasts were collected by centrifugation at 3500 g for 5 min and resuspended in 3.5 ml of 15% ficoll buffer (15% ficoll, 0.2 M sorbitol, 10 mM PIPES-KOH, pH 6.8). 3.5 μl of DEAE-Dextran (50 mg/ml) was added to the spheroplasts, and the sample was incubated for 3 min on ice and then for 5 min at 30 °C. 3.5 μl of MgCl₂ (1.5 M) was added to the lysate to terminate the reaction. The lysate was transferred to 13PA tubes (Koki Holdings, Japan) and overlaid with 3 ml of 8% Ficoll, 4 ml of 4% Ficoll, and 1 ml of 0% Ficoll. The tubes were centrifuged at 110,000 g for 90 min. The vacuolar fraction was collected from the 0/4% interphase, and protein concentrations from vacuoles were determined by a BCA protein assay reagent kit (TransGen, China). Samples were digested by concentrated nitric acid for Fe determination by ICP-MS (Agilent 7900, USA).

To generate PM-targeted proteins, the PM signal peptide ENO2(169) was amplified from yeast DNA and inserted into *pYES2-GmNRAMP2a/2b* vectors. The primers are shown in Supplementary

Data 4. The recombinant vectors *pYES2-ENO2(169)-GmNRAMP2a/2b* were transformed into Fe uptake defective mutant *fet3fet4*[59]. After selected by SD-Ura with glucose, yeast cells were cultured by YNB (-Fe) medium with yeast synthetic Drop-out medium supplements (-Ura) and glucose to exponential phase, and then were spotted onto SD-Ura plates with galactose and different concentrations of $FeCl_3$. For yeast cell density determination, yeast cell suspensions were diluted to an $OD_{600}$ of 0.1, and then incubated with galactose and different concentrations of $FeCl_3$ at 30 °C for 21 h. The values of $OD_{600}$ were dynamically determined. For short-term $^{57}Fe$ uptake, yeast cells were cultured by YNB (-Fe) medium with yeast synthetic Drop-out medium supplements (-Ura) and galactose to exponential phase, and collected by centrifugation at 3000 g for 5 min. The cells were washed twice with sterile water and incubated with 1, 5, 10 or 100 μM $^{57}FeCl_2$ (96.1% $^{57}Fe$; Trace Sciences International, Canada) for 5 min at RT. Yeast cells were collected and digested by concentrated nitric acid for $^{57}Fe$ determination by ICP-MS using stable isotope mode (Agilent 7900, USA).

## Separation of infected and uninfected cells
Intact infected and uninfected cell was isolated according to ref. 60 with some modifications. The fixation zone of mature root nodules (1-2 g) was dug out and cut into pieces, and then incubated in 5 mL enzyme solution (1% cellulase R-10, 0.1% pectolyase Y-23, 0.6 M mannitol, 10 mM MES-KOH (pH 5.7), 1 mM $MgCl_2$, 0.5% BSA, 0.5% dextran sulfate) at 28°C for 30 min with gentle shaking (40 rpm). Samples were filtered through three-layer tea bag to remove small tissue debris and bacteroids. The residues on the tea bag were then collected and washed three time with the same solution without enzyme. The cleaned samples were further incubated in 10 mL enzyme solution at 28°C without shaking for 2 h, and followed with occasional shaking for 1 h. The samples were filtered by three-layer tea bag, and followed by 30 μm nylon mesh. Cells on the mesh were suspended in 20 mM MOPS-KOH (pH 7.5) containing 0.6 M mannitol and 5 mM $CaCl_2$ on ice. Infected cells (larger, irregular-shaped and reddish-brown color) and uninfected cells (smaller, regular-shaped and nearly transparent) were separated by glass capillary tubes under a microscope. After quantified by using a hemocytometer, the collected cells were dried and digested by concentrated nitric acid for Fe determination by ICP-MS (Agilent 7900, USA).

## LA-ICP-TOF-MS
Nodules were first embedded in resin and then sliced into 10 μm thick sections according to the method mentioned in Perls/DAB staining. These sections were analyzed using a LA unit (NWR 193ImageGEO; New Wave Research) with the following settings: energy: 1 J/cm²; scan rate: 16000 μm s⁻¹; ablation frequency: 200 Hz; spot size: 8 × μm. Element signals were obtained using TOF-ICP-MS (TOF-WERK, Switzerland) with the following settings: Vendor: Tofwerk; Type: icpTOF R; Nebulizer gas flow: 1 L min⁻¹; RF power: 1400 W; Detector: MCP; Dwell time: 5 ms. All element signals were normalized to $^{13}C$ and converted to element images using iolite 4 software (http://iolite-software.com/). Three biological replicates were tested. This experiment was performed by Shanghai Chemlabpro Technology Co., Ltd.

## Transcriptional inhibition by GmNIGTs or GmNINs/NLPs
For segmental construction of *GmNRAMP2b* promoters, the *GmNRAMP2b* upstream regions of 2.5-kb, 2-kb, 1.5-kb, 1-kb, and 0.5-kb were amplified and cloned into the pFGC5941-GUS vector containing *p35S: GFP* cassette, respectively. The constructed vectors were transformed into hairy roots. Transgenic hairy roots from 25-d-old seedlings were inoculated with rhizobia and grown in low-N solution for 17 d, and then treated with or without H-N solution for 1 d. Nodule samples were collected for RNA extraction and gene expression analysis.

For transcriptional repressor activity assays, *6×GAL4-TATA* sequence was synthesized and inserted into pFGC5941-p35S-GFP vector to construct the reporter plasmid. Each NIGT gene was amplified and fused with GAL4 DNA binding domain (BD) at its N terminus to construct the effector plasmid. pFGC5941-p35S-RFP vector was used as internal control. The reporter, effector and internal control vectors were co-transformed into tobacco leaf, and GFP / RFP fluorescence signals were detected after 2-d incubation. In parallel, the reporter and effector were transformed into hairy roots, and nodules at 17 dpi from hairy roots were used for gene expression analysis.

To construct the overexpression lines, the ORFs of *GmNINs/ GmNLPs* were amplified and individually inserted into the *AscI* and *XbaI* sites of pFGC5941-*p35S* construct. The constructed vectors were transformed into hairy roots, and nodules at 17 dpi from hairy roots were used for gene expression and phenotypic analysis.

## Yeast one-hybrid assay
For yeast one-hybrid assay, the ORFs of *GmNIGT1a&1b*, *GmNIN1b* and *AtNLP1* were amplified and cloned in frame after transcriptional activation domain (AD) of GAL4 transcription factor in pB42AD respectively, which were used as effectors. The −500 bp promoter regions of *GmNRAMP2a&2b*, $4 \times NRE_{AtNIR1}$, $4 \times NRE_{GmNRAMP2a}$, $4 \times NRE_{GmNRAMP2b}$ and $6 \times NIE$ were cloned into the upstream of the *lacZ* reporter gene in pLacZi vector respectively, which were used as reporters. The effectors combined with the reporters were introduced into yeast strain EGY48 and cultured on SD medium (-Trp-Ura) containing X-gal at 30 °C. After 3 days, the yeast growth was photographed.

## EMSA assay
The purified GST-NIGT1a/1b and the oligonucleotides described in Fig. 6h were used for EMSA. To perform the EMSA, Oligonucleotides were end labeled with or without (competitor) Cy5 as probes. The coding sequence of NIGT1a/1b was individually introduced into PEGX4T-1. GST-NIGT1a/1b constructs and empty GST vectors were introduced into the *E. coli* strain DE3 to induce protein expression. The induced proteins were purified with Glutathione Sepharose 4B and then eluted with 10 mM glutathione. The Cy5-labelled probe (500 nM) was incubated with 2 μg recombinant protein in a reaction (100 mM Tris-HCl [pH 7.5], 100 mM KCl, 50 mM $MgCl_2$, 2.5 mM DTT) for 30 min at 4 °C. For competition assays, unlabeled double-stranded DNA was added to the binding reaction. The EMSA reactions were subjected to electrophoresis on 3.5% polyacrylamide gels in 0.5×Tris borate EDTA (TBE) buffer at 4 °C in the dark. Electrophoresis was performed at 100 V for 60 min. The fluorescence measurement of the polyacrylamide gel was detected on a LICOR Odyssey CLx system at 635 nm for excitation and 700 nm for emission.

## In situ overexpression of *GmNRAMP2b* and *GmNIGT1a*
To construct the overexpression lines, the 2.5 kb promoter sequences of *GmNramp2b* with the ORF of *GmNIGT1a* or the 2.4 kb promoter sequences of *GmNIGT1a* with the ORF of *GmNramp2b* were amplified and individually inserted into the *EcoRI* and *AscI* sites of pFGC5941-GFP construct. These constructs were transformed into *Agrobacterium rhizogenes* strain K599 for hairy-root transformation. Transformed hairy roots from 25-d-old seedlings were inoculated with an RFP-tagged rhizobium. Transgenic nodules at 20 dpi were transplanted to low-N (CK) or H−N for 2 d, and then were used for immunostaining and Perls/DAB staining.

## Statistics & Reproducibility
Statistical analyses were performed using GraphPad software. Means were compared using One-way ANOVA (Tukey-test) or unpaired two-sided *t*-test. Sample sizes were chosen based on our experience on the experimental variability of this type of experiment and the desire to

get statistically significant data to support meaningful conclusions. The number of independent biological seedlings or replicates has been shown in each figure legend. No data were excluded. Each experiment was repeated at least two times, and similar results were obtained. Seedlings were grown randomly in the growth chamber. Experiments were not blinded. Data were always collected according to the genotype of plants.

## Reporting summary
Further information on research design is available in the Nature Portfolio Reporting Summary linked to this article.

## Data availability
The authors declare that the data supporting the findings of this study are available within the paper and its supplementary information files. RNA-seq data have been deposited at NCBI (National Center for Biotechnology Information, project accession number PRJNA875247). Source data are provided with this paper.

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

## Acknowledgements

We thank Dr. Yong-Jia Zhong for providing RFP-expressing rhizobia, Dr. Jian-Feng Ma for providing *fet3fet4* mutant and stable isotope $^{57}$Fe, Dr. Xi Chen for providing *Δccc1* mutant, Dr. Wen-Fei Wang for providing Nodulin 26 antibody. This work is financially supported by the National Natural Science Foundation of China (No. 32370284).

## Author contributions

Z.C. conceived and designed the experiments. M.Z. and Y.L. performed most of the experiments. X.Y. and Y.G. helped with the construction of the CRISPR-Cas9 materials. J.Z., S.L., and H.C. contributed to vector constructions. S.B. contributed to symbiosome isolation. C.C. performed tobacco transient expression assay. D.Z. and A.X. performed yeast vacuolar isolation. J.L. and Q.M. performed the immunoelectron microscopy. Y.Z. and D.D. helped with the separation of infected and uninfected cells. M.Z., Y.L., and Z.C. analyzed the data. Z.C. helped with microscopic observation and wrote the manuscript. All authors discussed the results and commented on the manuscript.

## Competing interests

The authors declare no competing interests.
