## [Peer Review File · Nature Communications]

REVIEWER COMMENTS

Reviewer #1 (Remarks to the Author):

This is an interesting manuscript identifying a new route for iron to reach infected cells in nodules via the uninfected cells and mediated by two NRAMP transporters (although the authors conclude it is not the only route). It also identifies a link between the response of nodules to high nitrogen conditions and iron homeostasis. The work is carefully executed with a lot of detailed analysis. The authors show the reduction in iron in nodules over time as they are exposed to high N and that addition of supplementary iron reverses this to some extent. Through transcriptome analysis they identify two NRAMP transporters that are downregulated in the high N treatment but upregulated in conditions of Fe deficiency. They investigate the expression of the two NRAMP2 genes and localise the protein products on the tonoplast of uninfected cells in young nodules and show that they both transport iron in yeast. When these genes are silenced the response to high N is less marked than in control plants suggesting that the disruption in SNF in high N is in part a result of the action of these transporters in uninfected nodule cells. They conclude that in the high N treatment iron export from the vacuoles of uninfected cells is inhibited blocking supply of iron to infected cells and so to the bacteroids. In Fe deficiency they suggest upregulation of the NRAMP2 transporters would increase supply of Fe to infected cells. They also identify transcription factors of the NIGT family that negatively regulate the two NRAMP genes in response to high N and assess their role via silencing and overexpression.

Major comments:

1. While I don't argue with most of the conclusions, a problem with the analysis is that much of the data is collected in immature nodules between 13 and 17 dpi. It is possible that N-fixation may not have begun at this point of nodule development and the authors show no measure of nitrogenase activity or nitrogen export to establish this. Most studies in soybean consider nodules mature and N-fixation at its peak around 28-32 dpi nodules. The analysis for the transcriptomics (Fig 2a, b), cellular expression (promoter-GFP) (Fig 3a-d) and localisation (promoter-CDS-GFP) (Fig3i, j) is therefore likely to miss any role for NRAMP2a and b in infected cells with fully developed (N-fixing) symbiosomes. It should be repeated (at least cellular expression and localisation of the protein product) in mature nodules- particularly as the expression of NRAMP2b is highest at 40 dpi (it increases 4-6 fold between 21 dpi and 30-50 dpi). Any discussion of function needs to include data collected when expression is highest and nodules are mature. This is particularly important because, as the authors mention, a paper published in 2003 (Kaiser et al. ref 23 in the manuscript) on DMT1 (NRAMP2b) shows localisation of the protein on the symbiosome membrane in nodules that are 28 dpi. If this is correct there may be a role for NRAMP2b in infected cells (I agree it is unlikely to be import of iron to symbiosomes) as well as its role in uninfected cells and this needs to be considered when analysing the NRAMP2 mutants and overexpression data. The localisation data in Kaiser et al. (ref 23) is convincing, even though it is biased to infected cells (by isolating

symbiosome membrane and focussing the immunolocalization on infected cells) and if the authors of this manuscript suggest a different cellular expression pattern for the gene and localisation for the protein product of NRAMP2b they should provide convincing data that the symbiosome membrane localisation of DMT1 is not correct. If NRAMP2b was present of the SM in infected cells it might be regulated differently to the protein in uninfected cells given NIGT1a, b are not expressed in infected cells.

2. While the identified NIGTs are important in the response of nodules to high N and they do regulate NRAMP2a and b in response to this, this does not mean that NRAMP2a and b are the major players regulated by these transcription factors. What other genes are regulated by NIGT1a and 1b? A recent publication, Wang et al. (Nature Comms 2023,14:4711) looked at responses to high N in mature nodules and described a transcriptional network regulated by a set of NAC transcription factors that overlapped with senescence. Are NIGT1a and b and the NRAMP2s included in this? Are there differences in response early and late in nodule development? The authors should integrate their results with those of other studies on high N treatment of nodules. This would put the role of NRAMP2a, b, NIGT1a, b into a wider context and improve the discussion.

3. Since NRAMP2a and b expression responds to iron deficiency would they eventually be upregulated as iron levels are reduced in the infected region of nodule. A timecourse of expression equivalent to the timeseries shown in Figure 1 could be presented to understand the activity of these genes.

4. I would expect iron to be withdrawn and to eventually be reallocated to the rest of the plant after high N treatment and that is what seems to be occurring in Figure 1. Can the authors comment of whether their transcriptomic data suggest a mechanism for this if NRAMP2a,b are not involved (since they are down regulated).

Other points:

Line 43. Be consistent in terminology. Symbiosome and peribacteroid unit are used and I assume refer to the same thing. Symbiosome (304 hits in NCBI) is a more commonly accepted term than peribacteroid unit (5 hits including 4 articles using the term PBM).

Line 58. Refs 20-22 did not suggest DMT1 had the same role as VIT1. Ref 23 suggested that DMT1 was responsible for iron transport into symbiosomes. Correct this. Change to “A previous report suggested that GmNRAMP2b (also known as GmDMT1) facilitated Fe transport from the cytoplasm of infected cells into symbiosomes [23] functioning in a similar manner to VIT-like (VTL) transporters [20-22].”

The figure legends lack detail and as methods are in the supplementary data it is hard to quickly interpret figures.

Figure 1 appears to represent mature nodules but the age of the nodules is not mentioned. How many days after infection are the nodules at 0 days in Figure 1a? This is relevant as the transcriptome analysis was done on immature nodules (13 dpi). Information on the age of the nodules should be presented in all figure legends.

In figure 1 what was the effect on SNF (measured by acetylene reduction or N export) over the timecourse in Figure 1a?

The transcriptomic analysis summarised in figure 2a, b and c are interesting but lack detail. For a manuscript published in this journal I would expect to be able to access supplementary files with all the transcriptomic data for the H-N and -Fe treatments. What were the 142 transcripts that overlapped the analysis. What other transcripts responded to the H-N treatment?

GmNRAMP2a and 2b are expressed in most tissues but expression is low in roots (see transcriptome data at <https://soyatlas.venanciogroup.uenf.br/> and <http://soybeancellatlas.org/>) so figure 2e is a bit misleading. Perhaps they could present the Phytozome expression data for NRAMP2a and b in the same way as they do the data for NIGT1a and b. It would be good to at least comment on the fact that based on their expression pattern these transporters probably play an analogous role to AtNRAMP3 and 4 in other plant tissues.

The phylogenetic tree in Figure 2d has AtNRAMP3 and 4 in different clades when in fact they are most closely related. Without the gene numbers (At, Glyma codes) it is hard to interpret the tree. Could the codes be put either in the figure legend or the methods section.

How long was the H-N treatment in figure 2 g and what age were the nodules for all nutrient deficiency treatments.

In Figure 3 the promoter -GFP analysis is done on nodules 17 dpi but the expression of the gene in nodules is not enhanced until after 21 dpi (see Figure 2f)

The available single cell data (Liu et al. 2023 Nature Plants 9: 515–524; Tabula Glycine v2.0 <http://soybeancellatlas.org/>) for NRAMP2a and b and NIGT1a and b should be incorporated into the analysis for these genes.

Figure 5f and h are not described in the figure legend for the figure. How is Fe intensity in infected zone measure- is 5f stained with Perls/DAB?

The cellular expression patterns of NRAMP2a,b and NIGT1a,b were assessed using a promoter-GFP construct and anti-GFP antibodies. The results are much less obvious than a promoter-GUS assay would be. They made GUS constructs for NRAMP2b but did not show images of the GUS staining of the nodules. It would be good to see the results for these constructs in sections of nodule sections mature nodules when expression of NRAMP2b is high.

Sup data line 44 – were gene with fold change larger than 2 or log2fold larger than 2 selected. Please correct.

Line 237: This statement “Prior to VTL, GmNRAMP2b (also known as GmDMT1) has long been considered to be the Fe transporter localized on the symbiosome membrane (SM) of soybean nodules” is not exactly correct as many publications have suggested that it is unlikely that DMT1 would export Fe out of the cell- see Gonzalez-Guerrero et al. 2014 (Frontiers Plant Sci 5:45), Brear et al. 2013 (Frontiers Plant Sci 4:359), Gonzalez et al. 2016 (Frontiers Plant Sci 7:1008), Clarke et al. 2014 (Frontiers Plant Sci 5:699).

Line 245: change “whenever” to “whether”

Line 283: Reword “This suggests that GmNRAMP2a&2b seem not simply act as an equal-redundant manner.” It is not clear.

Line 284: change to “when large amounts of Fe are required in nodules”

What happens to leghaemoglobin in the nramp2ab nodules?

Reviewer #2 (Remarks to the Author):

NCOMMS-23-48265 describes both the functional and regulatory control of two Fe transport proteins GmNRAMP2a and GmNRAMP2b found expressed in soybean root nodules. Fe is an essential element used to support the development of the symbiotic partnership between plant and rhizobia and the metabolic activities in the nodule that lead to symbiotic nitrogen fixation by the partnering internally localised rhizobia bacteria. NRAMP proteins are one of multiple classes of described Fe transport proteins identified in plants (NRAMP, VIT, YSL, ZIP and MATE) that collectively contribute to Fe homeostasis in plant cells/tissues at various levels of responsibility and developmental time periods. NRAMP2b has been previously identified and characterised two decades earlier as a Fe transport protein (DMT1) in soybean nodules with a localisation identified at the symbiosome membrane.

The work presented improves our understanding of NRAMP (DMT1) activity in soybean nodules, which is highlighted by the described regulatory control through N supply and Fe availability, two areas that were not well understood previously. Exogenous N supply appears to disrupt NRAMP expression, which then disrupts Fe delivery to infected cells from reservoirs of stored Fe in uninfected cells sitting next to infected cells where invaded rhizobia (bacteroids) are housed in symbiosome structures. This brings forward a new interpretation between the role of infected and uninfected cells in nodule functionality and co-dependence for symbiotic outcomes. The manuscript has also described and shown evidence of a negative N-dependent regulatory network of two genes (NIG1a,1b) that influence transcriptional activity of NRAMP promoters. The proposed hypothesis focusses on Fe availability as a key control point that regulates N₂-fixation through a N-dependent homeostasis signalling cascade. Further work is required to link the broader Fe homeostatic systems operating nodules that support N₂-fixation.

Clarification and possible revision:

1) To start, the methodology and legend descriptions require significant upgrades to allow the reader to follow the research and interpret the outcomes. I suggest re-writing the methods section so it follows synchronously with the results presented in the manuscript. This will provide experimental context to the data that is being presented and allow those not familiar with this symbiotic process to better understand the subtleties of its regulation across its developmental life cycle. Inclusions of tissue age, type, state and treatment description will be helpful to the reader and reviewer.

2) This manuscript follows on other reports regarding Fe transport in legume nodules, including the description of VTL1 (Liu et al 2020, Brear et al 2020 and Walton et al 2020) and YSL7 (Wu et al., 2022). With each of these prior manuscripts, the relationship of Fe supply to symbiotic bacteroids has been described or speculated. Unfortunately, there is no information on the role of VTL1 or YSL7 relative to the two NRAMP proteins being examined (except in Figure 4b, examining Fe transport into vacuoles). For example, loss of *vtl1-1* disrupts infected region Fe levels quite dramatically (Liu et al), while the loss of NRAMP2b or its down regulation by high N has a more modest impact on infected cell Fe content. I would have expected that if uninfected cells are a primary mechanism for Fe delivery to infected cells (through plasmodesmata based connections or apoplast based pathways as highlighted in Figure 7), Fe would still flow from the uninfected cells via NRAMP2b in the *vtl1-1* mutant and infected cell Fe content remained high? Have the authors looked at Lb content, Lb-Fe management in the infected cells with the loss of *vtl1-1* or NRAMP2b as a mediator or buffer for Fe content in the infected cells, Fe in the infected cells is important for both Lb and nitrogenase function. It would appear from the previous *vtl-1* study that Fe content in vascular bundles, outer cortical cells and uninfected cells remains high. It is important that this study identifies the link between NRAMP and the other Fe intermediary proteins operating in nodules to provide the best conclusions regarding Fe homeostasis and N₂-fixation activity. It would be useful for the authors to comment and/or review the activities of the other Fe homeostatic systems which would be operating in the nodule?

3) The localisation of NRAMP2b and 2A has been suggested to be primarily tonoplast localised inside uninfected nodule cells. The evidence for this is strong. To rule out previous identities on the symbiosome membrane will require further evidence using immunogold labelling and TEM as well as western blot analysis of purified PM, TM, and symbiosome membrane fractions. It is quite clear strong localisation of GFP tagged protein is evident in uninfected cells, which has a dominate vacuole relative to the dispersed vacuole found in infected cells mainly replaced by thousands of small symbiosomes. However, there is an underlying faint signal (Figure 3b (anti-GFP) which could be explained by a low level of expression or distribution across many symbiosomes. I would suggest images be developed in matured nodules exposed to a low Fe treatment be used that would mirror the expression pattern of NRAMP2b? All of the anti-GFP tissue images are with +Fe and young nodules (17 DPI). This is the lowest expression level of NRAMP2b as shown in Figure 2f.

4) The Fe transport studies should include flux analysis from whole cells, non PM re-directed constructs (non ENO2(169)) and the use of Fe⁵⁵ and short-term influx experiments to quantify activities. 2-hour incubations and ICP-MS analysis does not allow for unidirectional influx measurements but represents net flux that is subject to both influx and efflux transport activities. Prior characterisation of NRAMP2b has shown transport properties associated with Fe transport and minor complementation of the *fet3fet4* mutant without the need for redirection to other cellular membranes.

5) Line 135-136 suggests GmNRAMP2a&2b are Fe influx transporters in yeast. Is this influx into vacuoles? If so can a Fe distribution analysis be completed in yeast cells expressing NRAMP2b to confirm this function?

6) Fig 5f highlights ultrastructural cell changes with \pm HN in WT and *nramp2ab* mutants. Its not clear if this is a Fe intensity image or not as the legend doesn't explain 5f.

7) Fig 5h/5n. I don't understand the comment that the nramp2a2b double mutant has a lower number of infected cells having lytic vacuoles re-appearing in the cells. The image would suggest the opposite to the quantified data?

Reviewer #3 (Remarks to the Author):

This paper shows that in the soybean-rhizobium symbiosis, inhibition of iron transport via NRAMP2a/b is associated with inhibition of symbiotic nitrogen fixation by externally supplied nitrogen nutrition. The authors show that NRAMP2 is localized in non-infected cells and regulates iron transport to infected cells, and that double mutants of NRAMP2a/b have abnormal iron transport to infected cells, resulting in reduced nitrogen fixation. In addition, they show that a GARP-type transcription factor, NIGT1, is responsible for the repression of NRAMP2a/b expression by nitrogen nutrition. The manuscript is generally well-written, and the data showing the function of NRSAMP2 is compelling. However, several points require further validation.

Comments:

Nitrate and ammonia are both known to inhibit root nodule symbiosis, but they are thought to act in different ways. Although NH₄NO₃ is used as the N source in this paper, whether nitrate or ammonia is actually more effective in controlling NRAMP2a/b should be examined for several important results.

I do not agree with L172-174 and 182-183. The mutants appear to be affected by H-N to the same extent as WT.

Fig. 3k: It is difficult to understand the effect of H-N on protein localization; why not examine vascular bundles where NRAMP2 is highly localized?

Data showing regulation of NRAMP2 by NIGT1 are scarce and not convincing; the relationship between the interaction and gene expression in vivo and in vitro should be investigated. Functional analysis of NIGT1 is also insufficient. As with NRAMP2, analysis needs to be performed using stable knockout plants.

There may be the impression that the statistics are done arbitrarily. In some graphs, only the results of a comparison of two conditions are shown, even though multiple data are shown in the same graph.

The overexpression experiment uses the 35S promoter, yet no explanation is given for the fact that the effects of overexpression are nodule-specific.

In each analysis, it is sometimes unknown whether stable transformants or hairy roots are used.

REVIEWER COMMENTS

Reviewer #1 (Remarks to the Author):

This is an interesting manuscript identifying a new route for iron to reach infected cells in nodules via the uninfected cells and mediated by two NRAMP transporters (although the authors conclude it is not the only route). It also identifies a link between the response of nodules to high nitrogen conditions and iron homeostasis. The work is carefully executed with a lot of detailed analysis. The authors show the reduction in iron in nodules over time as they are exposed to high N and that addition of supplementary iron reverses this to some extent. Through transcriptome analysis they identify two NRAMP transporters that are downregulated in the high N treatment but upregulated in conditions of Fe deficiency. They investigate the expression of the two NRAMP2 genes and localise the protein products on the tonoplast of uninfected cells in young nodules and show that they both transport iron in yeast. When these genes are silenced the response to high N is less marked than in control plants suggesting that the disruption in SNF in high N is in part a result of the action of these transporters in uninfected nodule cells. They conclude that in the high N treatment iron export from the vacuoles of uninfected cells is inhibited blocking supply of iron to infected cells and so to the bacteroids. In Fe deficiency they suggest upregulation of the NRAMP2 transporters would increase supply of Fe to infected cells. They also identify transcription factors of the NIGT family that negatively regulate the two NRAMP genes in response to high N and assess their role via silencing and overexpression.

Answer: Thank you very much for your positive comments.

Major comments:

1. While I don't argue with most of the conclusions, a problem with the analysis is that much of the data is collected in immature nodules between 13 and 17 dpi. It is possible that N-fixation may not have begun at this point of nodule development and the authors show no measure of nitrogenase activity or nitrogen export to establish this. Most studies in soybean consider nodules mature and N-fixation at its peak around 28-32 dpi nodules. The analysis for the transcriptomics (Fig 2a, b),

cellular expression (promoter-GFP) (Fig 3a-d) and localisation (promoter-CDS-GFP) (Fig3i, j) is therefore likely to miss any role for NRAMP2a and b in infected cells with fully developed (N-fixing) symbiosomes. It should be repeated (at least cellular expression and localisation of the protein product) in mature nodules- particularly as the expression of NRAMP2b is highest at 40 dpi (it increases 4-6 fold between 21 dpi and 30-50 dpi). Any discussion of function needs to include data collected when expression is highest and nodules are mature. This is particularly important because, as the authors mention, a paper published in 2003 (Kaiser et al. ref 23 in the manuscript) on DMT1 (NRAMP2b) shows localisation of the protein on the symbiosome membrane in nodules that are 28 dpi. If this is correct there may be a role for NRAMP2b in infected cells (I agree it is unlikely to be import of iron to symbiosomes) as well as its role in uninfected cells and this needs to be considered when analysing the NRAMP2 mutants and overexpression data. The localisation data in Kaiser et al. (ref 23) is convincing, even though it is biased to infected cells (by isolating symbiosome membrane and focussing the immunolocalization on infected cells) and if the authors of this manuscript suggest a different cellular expression pattern for the gene and localisation for the protein product of NRAMP2b they should provide convincing data that the symbiosome membrane localisation of DMT1 is not correct. If NRAMP2b was present of the SM in infected cells it might be regulated differently to the protein in uninfected cells given NIGT1a, b are not expressed in infected cells.

Answer: Thank you for your valuable suggestions. In response to your recommendations, we have conducted an extended time-course study (17, 24, 30, 40 dpi) of the gene expression and protein localization of GmNRAMP2a/2b in nodules. We observed that, regardless of the time point, the gene expression pattern of *GmNRAMP2b* was exclusively in the uninfected cells and not in the infected cells. The subcellular localization of GmNRAMP2b was on the tonoplast of uninfected cells. Besides, *GmNRAMP2a* was little expressed throughout the entire period of nodule development but can be significantly induced by -Fe, and showed similar gene expression and protein localization patterns to GmNRAMP2b (data from 30 dpi were shown below). We have added these results into the revised version (Supplementary Fig. 3).

Furthermore, to clarify the presence of GmNRAMP2b (DMT1) protein on the symbiosome membrane (SM), we performed immunogold electron-microscopy using a specific GFP antibody. The results revealed that GmNRAMP2b is localized only on the tonoplast of uninfected cells, with no detection on the SM (as illustrated in the figures below, supplementary Fig. 6a). Furthermore, there was no any immunogold signals observed in WT, suggesting the specificity of GFP antibody.

The subcellular localization has been further confirmed by western blot analysis (data shown below, supplementary Fig. 6b). We started by separating the nodules into symbiosomes and the remaining nodule tissue. After that, we isolated the tonoplast and plasma membrane proteins from the remaining nodule tissue. We also separated the symbiosomes (S) into symbiosome membranes (SM), symbiosome space (SS), and bacteroids (B). From our results, it can be concluded that the GmNRAMP2b protein tagged with GFP displayed the same fractionation pattern as the V-type ATPase (a known tonoplast membrane marker protein), but differed from the pattern shown by H⁺-ATPase (a known plasma membrane marker protein). Furthermore, we did not detect the GFP-tagged GmNRAMP2b protein in any subcellular structure of the symbiosomes, whereas the Nodulin-26 protein (a marker protein for the SM) showed a specific localization on the SM. It is worth mentioning that Kaiser et al. (2003) used a polyclonal antibody to detect NRAMP2b (DMT1), but the specificity of this antibody in soybean nodules was not investigated, leaving it uncertain whether it binds to NRAMP2b. In contrast, we used an GFP-specific antibody and have conducted negative controls, showing no signal in the wild type. Thus, the results of immunogold labeling,

immunostaining and western blot assay in our study are more convincing than those of Kaiser et al. (2003).

Regarding the concern about using immature young nodules in our experiments, please allow us to explain why we chose these nodules for our studies. **Firstly**, the 4-d-old stable transgenic seedlings mentioned in our paper were inoculated with rhizobia and then cultured for 21-25 d. Our system utilizes hydroponics in a growth chamber, providing soybeans with ample nutrition and space. We have observed that plants grown in this system tend to be larger and healthier compared to those grown in vermiculite. We reason that this might be why nodules develop faster than normally expected in our study. We acknowledge that there was some mistakes in describing the age of the nodules in figure legends and methods; we apologize for this mistake, which has now been corrected. **Secondly**, the 17-19 dpi nodule samples used in the paper were from transgenic hairy roots. Typically, we inoculate hairy roots on soybean seedlings around 25-d-old. At this stage, due to the larger biomass of the shoot, nodules on hairy roots tend to grow relatively quickly. We have previously tested nitrogenase activity across different stages and found that by 17 dpi, the nodules had already reached the nitrogen-fixing ability of mature nodules (as shown in the figure below). **Thirdly**, it is true that transcriptomic and gene expression assays involved immature nodules in figure 2a-c, 2g (17 dpi). We harvested these nodules once metal deficiency symptoms were observed to avoid large transcriptional changes by side effects. At that time, we did not assess

whether the nodules were at the maturity stage. However, we have also examined the gene expression in mature nodules (Fig. 2h-i, Supplementary Fig. 2), and the gene responses to -Fe and H-N were consistent with these results from immature nodules; therefore, we decided to retain this data in the paper. We appreciate your understanding of using immature young nodules in our experiments.

References:

- Liu, Z. et al. Integrated single-nucleus and spatial transcriptomics captures transitional states in soybean nodule maturation. *Nature Plants* 9, 515-524 (2023).
- Luo, Y. et al. Quantitative proteomics reveals key pathways in the symbiotic interface and the likely extracellular property of soybean symbiosome. *J. Genet. Genomics* 50, 7-19 (2023).

2. While the identified NIGTs are important in the response of nodules to high N and they do regulate NRAMP2a and b in response to this, this does not mean that NRAMP2a and b are the major players regulated by these transcription factors. What other genes are regulated by NIGT1a and 1b? A recent publication, Wang et al. (*Nature Comms* 2023,14:4711) looked at responses to high N in mature nodules and described a transcriptional network regulated by a set of NAC transcription factors that overlapped with senescence. Are NIGT1a and b and the NRAMP2s included in this? Are there differences in response early and late in nodule development? The authors should integrate their results with those of other studies on high N treatment of nodules. This would put the role of NRAMP2a, b, NIGT1a, b into a wider context and improve the discussion.

Answer: Thank you for your insightful suggestions. In response to your comments, we have

conducted a comparative transcriptomic assay, which revealed 47 potentially NIGT1ab-regulated genes (as shown in Supplementary Dataset 3). These genes are up-regulated in *nigt1ab* double knockout line and down-regulated in *NIGT1a*-overexpressing line and under high-nitrogen (H-N) conditions. Interestingly, we found that, beyond *NRAMP2ab*, important iron-homeostasis-related genes such as *IMAs* and *BRUTUSs* are also under the regulation of NIGT1ab (as shown in the figure below, Supplementary Fig. 17).

Recent studies from a Japanese research group (Ito et al., Nature Communications, 2024) revealed the regulatory role of IMAs in the symbiosis and nitrogen fixation in the nodules of *Lotus japonicus*, further verifying the vital role of iron nutrition in nodules. We are currently investigating the function of the *IMA1234* genes in the soybean nodules and hope to publish these results in the near future.

Following your advice, we reviewed the Chip-seq data from Wang et al. (Nature Communications, 2023). Indeed, apart from NIGT1b, NIGT1a, 1c, and 1d were identified as direct targets of NACs (Supplementary Data 11 from Wang et al. 2023). Additionally, after consulting with the corresponding author of this paper and re-examining their raw data, we discovered that NIGT1b is also regulated by NACs, even though it was not included in the list due to its low threshold. This suggests that NACs could at least partially mediate the transmission of N signals through the NIGT1 signaling pathways. We have incorporated these findings into Discussion.

To investigate the response of *NIGT1ab* to nitrogen at different stages of nodule development, we

examined the gene expression of *NIGT1ab* at different time points (7, 14, 17, 21, 30, 40, 50 dpi) in nodules. We found that throughout the developmental process, the expression was significantly up-regulated by H-N (as shown in the figure below, Supplementary Fig. 2c-d). This indicates that the role of GmNIGT1a&1b in N perception is active throughout early to late nodule development.

References:

Ito, M. et al. IMA peptides regulate root nodulation and nitrogen homeostasis by providing iron according to internal nitrogen status. *Nat. Commun.* 15, 733 (2024).

Wang, X. et al. The NAC transcription factors SNAP1/2/3/4 are central regulators mediating high nitrogen responses in mature nodules of soybean. *Nat. Commun.* 14, 4711 (2023).

3. Since NRAMP2a and b expression responds to iron deficiency would they eventually be upregulated as iron levels are reduced in the infected region of nodule. A time course of expression equivalent to the timeseries shown in Figure 1 could be presented to understand the activity of these genes.

Answer: To address this issue, we performed a time-course analysis of gene responses to iron deficiency, high nitrogen levels, and simultaneous iron deficiency and high nitrogen. Our results clearly show that under both iron deficiency and high nitrogen conditions, the expression of *GmNRAMP2a* and *2b* remains suppressed (as shown in the figure below, Fig. 2h-i), suggesting that nitrogen signals play a dominant role in expression regulation. Therefore, we propose that in the soybean nodules, nitrogen signals regulate gene expression upstream of iron signals. This hypothesis can be supported by the results showing that NIGT1ab regulates the expression of *IMAs* and *BRUTUSs*, both of which are critical proteins involved in the regulation of iron signaling

pathways (Supplementary Fig. 17). This finding also suggests that the decrease in iron content in nodules under high nitrogen conditions represents an active regulatory strategy of the host to reduce nitrogen fixation in root nodules.

4. I would expect iron to be withdrawn and to eventually be reallocated to the rest of the plant after high N treatment and that is what seems to be occurring in Figure 1. Can the authors comment of whether their transcriptomic data suggest a mechanism for this if NRAMP2a,b are not involved (since they are down regulated).

Answer: Thank you for your suggestion, and we agree with your opinion. Previous reports have suggested that nutrients transported to nodules should be recycled back into the plant during nodule senescence (Van de Velde et al., 2006), although we currently lack evidence to demonstrate that iron is moved back to the roots or shoots under high nitrogen conditions.

Assuming that iron can be recycled by the plant, it is plausible that the transporters responsible for this iron transfer might be induced by high nitrogen and inhibited by iron deficiency. We found a pair of VIT-like proteins and four ferritin proteins that fit this expression pattern (Fig. 2c). However, considering their functional roles, these proteins are more likely involved in storing iron, thereby preventing Fe availability to rhizobia, similar to the role of NRAMP2, rather than transporting iron back to the host plant. Therefore, based on these findings, we can only conclude that the host plant has slowed or halted the transport of iron to the nodules, but we are unable to demonstrate that the iron from nodules is ultimately reallocated to the host plant under high-N conditions. The reduced concentration of iron in the fixation zone under high-N conditions (Fig. 1a) might be attributed to a

dilution effect resulting from the proliferation of rhizobia.

Reference:

Van de Velde, W. *et al.* Aging in legume symbiosis. A molecular view on nodule senescence in *Medicago truncatula*. *Plant Physiol* **141**, 711-720 (2006).

Other points:

Line 43. Be consistent in terminology. Symbiosome and peribacteroid unit are used and I assume refer to the same thing. Symbiosome (304 hits in NCBI) is a more commonly accepted term than peribacteroid unit (5 hits including 4 articles using the term PBM).

Answer: Thanks for the suggestions. We have revised all ‘PBU’ to ‘symbiosome’ in the revised text.

Line 58. Refs 20-22 did not suggest DMT1 had the same role as VIT1. Ref 23 suggested that DMT1 was responsible for iron transport into symbiosomes. Correct this. Change to “A previous report suggested that GmNRAMP2b (also known as GmDMT1) facilitated Fe transport from the cytoplasm of infected cells into symbiosomes [23] functioning in a similar manner to VIT-like (VTL) transporters [20-22].”

Answer: Thanks for the suggestions. This sentence has been revised accordingly.

The figure legends lack detail and as methods are in the supplementary data it is hard to quickly interpret figures.

Answer: We have added more details of the experimental methods in the figure legends.

Figure 1 appears to represent mature nodules but the age of the nodules is not mentioned. How many days after infection are the nodules at 0 days in Figure 1a? This is relevant as the transcriptome analysis was done on immature nodules (13 dpi). Information on the age of the nodules should be presented in all figure legends.

Answer: We deeply apologize for the oversight in the description of the nodule growth period. 4-d-

old seedlings were inoculated with rhizobia, and nodules at 21 dpi were used for high-N treatment for 1, 2, 3, or 4 d in Figure 1. We divide the seedlings into batches and subject them to high nitrogen treatment daily to ensure simultaneous sample collection (at 25 dpi). The information on the age of the nodules has been presented in all figure legends of revised text.

In figure 1 what was the effect on SNF (measured by acetylene reduction or N export) over the time course in Figure 1a?

Answer: The result of time-course N export rate of nodules has been added into figure 1c (shown below), which shows a gradually decreased N export rate after nodule transfer to H-N.

The transcriptomic analysis summarised in figure 2a, b and c are interesting but lack detail. I For a manuscript published in this journal I would expect to be able to access supplementary files with all the transcriptomic data for the H-N and -Fe treatments. What were the 142 transcripts that overlapped the analysis. What other transcripts responded to the H-N treatment?

Answer: Thanks for the suggestions. We have presented these analyzed transcriptional data in Supplementary Dataset 1. Besides, the raw data of RNA-seq has been uploaded to the NCBI.

GmNRAMP2a and 2b are expressed in most tissues but expression is low in roots (see transcriptome data at <https://soyatlas.venanciogroup.uenf.br/> and <http://soybeancellatlas.org/>) so figure 2e is a bit misleading. Perhaps they could present the Phytozome expression data for NRAMP2a and b in the same way as they do the data for NIGT1a and b. It would be good to at least comment on the fact that based on their expression pattern these transporters probably play an analogous role to

AtNRAMP3 and 4 in other plant tissues.

Answer: Based on your suggestion, we have shown a heatmap of *GmNRAMP2ab* expression pattern in organs/tissues (as shown in the figure below, Fig. 2d). In addition, we have added your comments into the Results: *GmNRAMP2a* and *2b* are expressed in most tissues, and only *GmNRAMP2b* is highly expressed in nodules. The expression of these two genes in other tissues may play a role similar to that of *AtNRAMP3* and *4* in Arabidopsis. Further investigations into the iron status and phenotypes of other tissues might be necessary to fully understand their functions.

The phylogenetic tree in Figure 2d has AtNRAMP3 and 4 in different clades when in fact they are most closely related. Without the gene numbers (At, Glyma codes) it is hard to interpret the tree. Could the codes be put either in the figure legend or the methods section.

Answer: Thank you for your checking. We have renamed the NRAMP genes according to the published papers. The gene numbers have been added to the phylogenetic tree, and the revised tree has been moved to the supplementary Fig. 1a.

Reference:

Qin. L., et al. Genome-wide identification and expression analysis of NRAMP family genes in soybean (*Glycine max* L.). *Frontiers in plant science* 8, 272851 (2017)

Peris-Peris. C., et al. Two NRAMP6 isoforms function as iron and manganese transporters and contribute to disease resistance in rice. *Molecular Plant-Microbe Interactions* 30, 385-398 (2017)

How long was the H-N treatment in figure 2 g and what age were the nodules for all nutrient deficiency treatments.

Answer: To investigate the expression in response to various nutrient stresses, seedlings grown in a low-N solution at 10 dpi were transferred to a low-N and Fe-, Mg-, Mo-, Mn-, Zn- or S-free solution for 7 d, or seedlings grown in a low-N solution at 16 dpi were treated with high-N for 1 d. We have added this information into figure legend.

In Figure 3 the promoter -GFP analysis is done on nodules 17 dpi but the expression of the gene in nodules is not enhanced until after 21 dpi (see Figure 2f).

Answer: The nodule samples for promoter -GFP analysis were from **transgenic hairy roots**. Typically, we inoculate hairy roots on soybean seedlings around 25 days old. At this stage, due to the larger biomass of the shoot, nodules on hairy roots tend to grow relatively quickly. We have previously tested nitrogenase activity across different stages and found that by 17 dpi, the nodules had already reached the nitrogen-fixing ability of mature nodules (Figure has been shown in page 6).

For time-course RT-qPCR analysis (original Figure 2f), 4-d-old seedlings were inoculated with

rhizobia and grown in a **vermiculate** medium supplied with low-N solution for 7, 14, 21, 30, and 50 days. We reason that this expression difference is due to the differences in nodule maturity, seedling growth stage as well as the growth medium between the two experiments. Considering that all experimental conditions in the paper were conducted under hydroponic conditions, we rechecked the time-course expression pattern using seedlings grown under **hydroponic culture** (figure shown below). The results from the hydroponically grown seedlings indicate that the expression of *NRAMP2b* increases with nodule development, reaching its peak at 21 days, which is much earlier than that grown in vermiculate. We speculate that under hydroponic conditions, the root system has sufficient space and nutrients to achieve its optimal growth state, compared to that grown in vermiculite. We have replaced Figure 2f with the new results.

The available single cell data (Liu et al. 2023 Nature Plants 9: 515–524; Tabula Glycine v2.0 <http://soybeancellatlas.org/>) for *NRAMP2a* and *b* and *NIGT1a* and *b* should be incorporated into the analysis for these genes.

Answer: Thank you for your good suggestions. We have now collected the results of these four genes from the two databases. The results from both databases consistently show that these four genes are enriched in the uninfected cells of the nodules. However, there are differences in cell classification between the two databases, which may be attributed to the use of different marker genes. In the Tabula Glycine database, *GmNIGT1a* and *Ib* are also enriched in the infection zone, where both infected and uninfected cells exist. The results are as follows:

From https://zhailab.bio.sustech.edu.cn/single_cell_soybean

From <http://soybeancellatlas.org/>

Additionally, we examined published soybean SM proteomics data (Luo et al., 2022) and did not detect NRAMP2a, 2b, NIGT1a, or 1b proteins.

Reference:

Luo, Y. et al. Quantitative proteomics reveals key pathways in the symbiotic interface and the likely extracellular property of soybean symbiosome. *J. Genet. Genomics* 50, 7-19 (2023).

Figure 5f and h are not described in the figure legend for the figure. How is Fe intensity in infected zone measure- is 5f stained with Perls/DAB?

Answer: We apologize for the missing information. Yes, it was stained with Perls/DAB. The detailed method is shown: Nodules were incubated overnight in fixation solution containing 50% (v/v) ethanol, 5% (v/v) glacial acetic acid, and 10% (v/v) formaldehyde solution and vacuum infiltrated for 30 minutes. The fixed nodules were dehydrated in a series of 50%, 60%, 70%, 80%, and 90% ethanol solutions, and then overnight dehydrated in 100% ethanol. Samples were embedded in Technovit 7100 resin (Kulzer) according to the kit instructions, and thin sections (2 μ m) were prepared. These sections were vacuum infiltrated for 15 min each with equal volumes of 4% (v/v) HCl and 4% (w/v) K-ferrocyanide (Perls stain solution), and incubated for 30 min at room temperature. For DAB intensification, fixed sections were washed with distilled water and incubated in a methanol solution containing 10 mM NaN₃ and 0.3% (v/v) H₂O₂ for 1 h. After washing with PBS, sections were then incubated in an intensification solution containing 0.025% (w/v) DAB, 0.005% (v/v) H₂O₂, 0.005% (w/v) CoCl₂ and 0.1 M PBS (pH 7.4) for 30 min prior to being washed with distilled water to stop the reaction. These sections were photographed using an optical microscope (Nikon Ni-U, Japan). Iron signal intensity was quantified using the grayscale difference analysis method in ImageJ software.

The cellular expression patterns of NRAMP2a,b and NIGT1a,b were assessed using a promoter-GFP construct and anti-GFP antibodies. The results are much less obvious than a promoter-GUS assay would be. They made GUS constructs for NRAMP2b but did not show images of the GUS staining of the nodules. It would be good to see the results for these constructs in sections of nodule sections mature nodules when expression of NRAMP2b is high.

Answer: Thank you for your suggestion. we have conducted GUS staining on transgenic nodules harboring *proGmNRAMP2a/2b-GUS* (at 17, 22 and 28 dpi). We observed that *proGmNRAMP2a-GUS* transgenic nodules did not exhibit GUS signal, whereas *proGmNRAMP2b-GUS* transgenic nodules showed strong GUS activity. Cross-section results revealed that the signal was mainly concentrated in the infection zone and the surrounding vascular bundles, consistent with their

expression patterns (as shown in the figure below). However, there are two disadvantages of GUS staining we encountered. First, the GUS signal can indeed be quantified; however, it is generally more challenging than quantifying signals from other reporter genes like GFP. We have attempted to quantify promoter activity using GUS signals, but the significant variability in the signals led to unreliable results. Consequently, we have alternatively employed quantitative PCR to assess the expression of the GUS gene, as shown in Supplementary Figures 13b. Second, the diffuse signals of GUS staining made it difficult to determine whether the signal was specifically localized to the uninfected regions of the nodule. This diffusion is attributed to the movement of the GUS reaction product, which can migrate from the site of enzyme activity, potentially leading to less precise localization of the staining. Hence, while GUS staining is a useful tool for indicating gene expression patterns, care must be taken in interpreting results due to the potential for signal diffusion. Considering these disadvantages of the GUS signal compared to immunostaining, we decided not to replace the results of immunostaining with the GUS results. We appreciate your understanding regarding this experiment.

Sup data line 44 – were gene with fold change larger than 2 or log₂fold larger than 2 selected. Please correct.

Answer: The gene with fold change (FC) larger than 2 (or log₂ FC >1) was selected. We have corrected this.

Line 237: This statement “Prior to VTL, GmNRAMP2b (also known as GmDMT1) has long been considered to be the Fe transporter localized on the symbiosome membrane (SM) of soybean

nodules” is not exactly correct as many publications have suggested that it is unlikely that DMT1 would export Fe out of the cell- see Gonzalez-Guerrero et al. 2014 (Frontiers Plant Sci 5:45), Brear et al. 2013 (Frontiers Plant Sci 4:359), Gonzalez et al. 2016 (Frontiers Plant Sci 7:1008), Clarke et al. 2014 (Frontiers Plant Sci 5:699).

Answer: Thanks. We have revised these sentences: Although previously GmNRAMP2b, also termed GmDMT1, was presumed to act as an Fe transporter at the symbiosome membrane (SM) of soybean nodules²³, this has been brought into question by researchers which suggest it is unlikely to be involved in exporting Fe out of cells^{34,35}. Given that the physiological role of GmNRAMP2b in SNF remains unconfirmed, its contribution to Fe homeostasis within nodules was not yet understood.

Line 245: change “whenever” to “whether”

Answer: We have changed it accordingly.

Line 283: Reword “This suggests that GmNRAMP2a&2b seem not simply act as an equal-redundant manner.” It is not clear.

Answer: We have reworded it into: This suggests that GmNRAMP2a&2b possess an asymmetrically redundant role in Fe transport.

Line 284: change to “when large amounts of Fe are required in nodules”

Answer: We have changed it accordingly.

What happens to leghaemoglobin in the *nramp2ab* nodules?

Answer: Thank you for your suggestions. We have employed transcriptomic analysis to investigate the expression of leghaemoglobin (Lb) in both wild-type plants and mutants of *nramp2ab* and *vtll-1* (the latter being defective in Fe transport into the symbiosome, as identified by Liu et al., 2020). Additionally, we have investigated the protein levels of Lb through western blot analysis (as shown in the figure below). Our findings reveal a slight reduction in the expression and protein levels of Lb in the *nramp2ab* mutants. However, this reduction is not as pronounced as that observed in the *vtll-1* mutants. This indicates that mutating these two genes does impact the homeostasis of Fe within nodules, which in turn affect Lb levels in the infected cells to varying extents.

Reviewer #2 (Remarks to the Author):

NCOMMS-23-48265 describes both the functional and regulatory control of two Fe transport proteins GmNRAMP2a and GmNRAMP2b found expressed in soybean root nodules. Fe is an essential element used to support the development of the symbiotic partnership between plant and rhizobia and the metabolic activities in the nodule that lead to symbiotic nitrogen fixation by the partnering internally localised rhizobia bacteria. NRAMP proteins are one of multiple classes of described Fe transport proteins identified in plants (NRAMP, VIT, YSL, ZIP and MATE) that collectively contribute to Fe homeostasis in plant cells/tissues at various levels of responsibility and developmental time periods. NRAMP2b has been previously identified and characterised two decades earlier as a Fe transport protein (DMT1) in soybean nodules with a localisation identified at the symbiosome membrane.

The work presented improves our understanding of NRAMP (DMT1) activity in soybean nodules, which is highlighted by the described regulatory control through N supply and Fe availability, two areas that were not well understood previously. Exogenous N supply appears to disrupt NRAMP expression, which then disrupts Fe delivery to infected cells from reservoirs of stored Fe in uninfected cells sitting next to infected cells where invaded rhizobia (bacteroids) are housed in symbiosome structures. This brings forward a new interpretation between the role of infected and uninfected cells in nodule functionality and co-dependence for symbiotic outcomes. The manuscript has also described and shown evidence of a negative N-dependent regulatory network of two genes (NIG1a,1b) that influence transcriptional activity of NRAMP promoters. The proposed hypothesis focusses on Fe availability as a key control point that regulates N₂-fixation through a N-dependent homeostasis signalling cascade. Further work is required to link the broader Fe homeostatic systems operating nodules that support N₂-fixation.

Thank you very much for your positive comments.

Clarification and possible revision:

1) To start, the methodology and legend descriptions require significant upgrades to allow the reader

to follow the research and interpret the outcomes. I suggest re-writing the methods section so it follows synchronously with the results presented in the manuscript. This will provide experimental context to the data that is being presented and allow those not familiar with this symbiotic process to better understand the subtleties of its regulation across its developmental life cycle. Inclusions of tissue age, type, state and treatment description will be helpful to the reader and reviewer.

Answer: Thank you for your suggestions. We have made extensive revisions to the Materials and Methods section, as well as the figure legends. This includes adjusting the order of methods, expanding the descriptions of the experiments, and enhancing the detailed descriptions of the experimental materials.

2) This manuscript follows on other reports regarding Fe transport in legume nodules, including the description of VTL1 (Liu et al 2020, Brear et al 2020 and Walton et al 2020) and YSL7 (Wu et al., 2022). With each of these prior manuscripts, the relationship of Fe supply to symbiotic bacteroids has been described or speculated. Unfortunately, there is no information on the role of VTL1 or YSL7 relative to the two NRAMP proteins being examined (except in Figure 4b, examining Fe transport into vacuoles). For example, loss of *vtl1-1* disrupts infected region Fe levels quite dramatically (Liu et al), while the loss of NRAMP2b or its down regulation by high N has a more modest impact on infected cell Fe content. I would have expected that if uninfected cells are a primary mechanism for Fe delivery to infected cells (through plasmodesmata based connections or apoplast based pathways as highlighted in Figure 7), Fe would still flow from the uninfected cells via NRAMP2b in the *vtl1-1* mutant and infected cell Fe content remained high? Have the authors looked at Lb content, Lb-Fe management in the infected cells with the loss of *vtl1-1* or NRAMP2b as a mediator or buffer for Fe content in the infected cells, Fe in the infected cells is important for both Lb and nitrogenase function. It would appear from the previous *vtl1-1* study that Fe content in vascular bundles, outer cortical cells and uninfected cells remains high. It is important that this study identifies the link between NRAMP and the other Fe intermediary proteins operating in nodules to provide the best conclusions regarding Fe homeostasis and N₂-fixation activity. It would be useful for the authors to comment and/or review the activities of the other Fe homeostatic systems which would be operating in the nodule?

Answer: We appreciate your constructive suggestions. In response to your questions, we have conducted transcriptome analysis and protein abundance analysis under control, high-nitrogen (H-N), and Fe-deficiency (-Fe) conditions for wild-type, *nramp2ab*, and *vtl1* mutants (results shown in the figure below, Supplementary Fig. 10 and 11).

Since ferritin is strongly correlated with intracellular iron levels (Briat et al., 2010), we first examined ferritin levels to reflect the state of iron in the nodules of the mutants. Interestingly, we observed a decrease in ferritin transcripts and protein levels in *nramp2ab*, whereas there was a noticeable increase in *vtl1* mutants. These results indicate that *nramp2ab* mutants are unable to transport iron from vacuoles to the cytoplasm, leading to reduced iron levels in cytoplasm, while *vtl1* mutants are impaired in transporting iron to symbiosomes, resulting in increased cytoplasmic iron levels. In parallel, the absence of both NRAMP2ab and VTL1ab led to a decrease in nitrogenase NifH levels and a reduction in leghemoglobin (Lb) at both the transcriptional and protein levels, suggesting that disruption of iron balance (whether due to reduction or excess cytoplasmic iron) diminishes the nitrogen-fixing capacity of nodules.

Additionally, we investigated the expression changes of iron-related transporter genes (*NRAMP*, *VIT*, *YSL*, *ZIP*, and *OPT*) in these mutants. We found that the *nramp2ab* mutation led to a decrease in *VTL1ab* expression, and the absence of *VTL1ab* not only decreased the expression levels of *NRAMP2ab*, but also many other putative iron transporter genes (including *GmYSL7*). These results suggest that maintaining iron homeostasis is a multifaceted process involving the synchronized operation of various transporters, and plants possess the adaptive ability to adjust the expression of iron-related transporters, to counterbalance disruptions in any single iron transport pathway. To further validate this hypothesis, we expressed *GmNRAMP2b-GFP* in *vtl1* mutants. We found that *GmNRAMP2b-GFP* was virtually undetectable in *vtl1-1* mutant, regardless of H-N treatment, which further confirms the existence of a feedback regulation system within nodules. We speculate that this system is designed to prevent excessive accumulation of iron in the nodule cytoplasm.

In summary, we believe VTL1 and NRAMP2 proteins play different roles in soybean nodules. VTL1 is a constitutively expressed, indispensable protein for maintaining the basic nitrogen-fixing capacity of infected cells, while NRAMP2 protein predominantly functions as a regulator in

uninfected cells, with its role becoming more pronounced when the nodule has a higher demand for iron. It was known that that the overall iron levels are low in *vtI1* mutants (Liu et al., 2020) because iron cannot enter symbiosomes, which are major iron pools in nodules, leading to a dramatic reduction in iron levels. However, this results in an excessive accumulation of iron in its cytoplasm (as indicated by the increase in ferritin levels). The excessive accumulation of iron in *vtI1a* induces feedback effects that affect the expression of other iron-related proteins, including *GmNRAMP2ab* and GmYSL7. This illustrates that the iron homeostasis mechanism in nodules is regulated by multiple iron transporters in coordination, to optimize the nitrogen-fixation capacity of the nodule.

References:

- Briat JF, Duc C, Ravet K, Gaymard F. 2010. Ferritins and iron storage in plants. *Biochimica et Biophysica Acta* 1800: 806–814.
- Liu S, Liao LL, Nie MM, Peng WT, Zhang MS, Lei JN, Zhong YJ, Liao H, Chen ZC (2020) A VIT-like transporter facilitates iron transport into nodule symbiosomes for nitrogen fixation in soybean. *The New phytologist* 226(5):1413-1428

3) The localisation of NRAMP2b and 2A has been suggested to be primarily tonoplast localised inside uninfected nodule cells. The evidence for this is strong. To rule out previous identities on the symbiosome membrane will require further evidence using immunogold labelling and TEM as well as western blot analysis of purified PM, TM, and symbiosome membrane fractions. It is quite clear strong localisation of GFP tagged protein is evident in uninfected cells, which has a dominate vacuole relative to the dispersed vacuole found in infected cells mainly replaced by thousands of small symbiosomes. However, there is an underlying faint signal (Figure 3b (anti-GFP) which could be explained by a low level of expression or distribution across many symbiosomes. I would suggest

images be developed in matured nodules exposed to a low Fe treatment be used that would mirror the expression pattern of NRAMP2b? All of the anti-GFP tissue images are with +Fe and young nodules (17 DPI). This is the lowest expression level of NRAMP2b as shown in Figure 2f.

Answer: Thank you for your valuable suggestions. To clarify the presence of NRAMP2b (DMT1) protein on the symbiosome membrane (SM), we performed immunogold electron-microscopy. The results revealed that NRAMP2b is localized only on the tonoplast of uninfected cells, with no detection on the SM (as illustrated in the figures below, Supplementary Fig. 6a). Kaiser et al. (2003) used a polyclonal antibody to detect NRAMP2b (DMT1), but the specificity of this antibody in soybean nodules was not investigated, leaving it uncertain whether it specifically binds to NRAMP2b. In contrast, we used a GFP-specific antibody and have conducted negative controls, showing no signal in the wild type. Thus, the results of immunogold labeling and immunostaining in our study are more convincing than those of Kaiser et al. (2003).

The subcellular localization has been further confirmed by western blot analysis (data shown below,

Supplementary Fig. 6b). We started by separating the nodules into symbiosomes and the remaining nodule tissue. After that, we isolated the tonoplast and plasma membrane proteins from the remaining nodule tissue. We also separated the symbiosomes (S) into symbiosome membranes (SM), symbiosome space (SS), and bacteroids (B, please refer to the following methods for detailed procedures). From our results, it can be concluded that the GmNRAMP2b protein tagged with GFP displayed the same fractionation pattern as the V-type ATPase (a known tonoplast membrane marker protein), but differed from the pattern shown by H⁺-ATPase (a known plasma membrane marker protein). Furthermore, we did not detect the GFP-tagged GmNRAMP2b protein in any subcellular structure of the symbiosomes, whereas the Nodulin-26 protein (a marker protein for the SM) showed a specific localization on the SM. Additionally, we examined published soybean SM proteomics data (Luo et al., 2022) and did not detect NRAMP2a, 2b, NIGT1a, or 1b proteins.

Besides, we have conducted an extended time-course study (17, 24, 30, 40 dpi) of the gene expression and protein localization of GmNRAMP2a/2b in nodules. We observed that, regardless of the time point, the gene expression pattern of GmNRAMP2b was dominantly in the uninfected cells. The subcellular localization of GmNRAMP2b was on the tonoplast of uninfected cells. Besides, GmNRAMP2a was little expressed throughout the entire period of nodule development but can be significantly induced by -Fe, and showed similar gene expression and protein localization patterns to GmNRAMP2b (data from 30 dpi with -Fe and +Fe conditions were shown below, Supplementary Fig. 3). It is worth mention that the faint signals in Fig. 3a-b might originates from background fluorescence, as we observed signals of equal intensity in the nodules from wild type.

4) The Fe transport studies should include flux analysis from whole cells, non PM re-directed constructs (non ENO2(169)) and the use of Fe⁵⁵ and short-term influx experiments to quantify activities. 2-hour incubations and ICP-MS analysis does not allow for unidirectional influx measurements but represents net flux that is subject to both influx and efflux transport activities. Prior characterisation of NRAMP2b has shown transport properties associated with Fe transport and minor complementation of the *fet3fet4* mutant without the need for redirection to other cellular membranes.

Answer: Thank you for your suggestion. Based on your suggestions, we have compared the short-term (5min) uptake of stable isotope ⁵⁷Fe (from Prof. Jian Feng Ma, Okayama University, Japan) among the *fet3fet4* yeast mutants expressing *GmNRAMP2a*, *GmNRAMP2b*, *ENO2(169)-GmNRAMP2a*, and *ENO2(169)-GmNRAMP2b* genes. We observed that only the expression of

ENO2(169)-GmNRAMP2a and *ENO2(169)-GmNRAMP2b* enhanced the short-time uptake of iron, while the expression of *GmNRAMP2a* and *GmNRAMP2b* did not (data shown below).

We recognized GmNRAMP2b as a potential Fe influx transporter given its capability to rescue the growth of the *fet3fet4* mutant, which is defective in ferrous Fe uptake as illustrated by Kaiser et al. 2003. This prompted us to hypothesize that the expression of GmNRAMP2b in yeast might enhance cellular Fe uptake. However, our previous findings indicate that the presence of GmNRAMP2b in yeast did not increase Fe uptake under both high and low Fe conditions (data not shown), and ⁵⁷Fe uptake experiment further confirm expression of GmNRAMP2b in yeast cannot enhance cellular Fe uptake. Furthermore, considering that GmNRAMP2b was found to be ONLY localized at the tonoplast of yeast cells (Supplementary Fig. 4a), we reason that the complementation of the *fet3fet4* strain by GmNRAMP2b from Kaiser et al. (2003) could be attributed to an indirect effect rather than an increased uptake of Fe by the yeast. This indirect effect might be due to GmNRAMP2b-mediate increment of Fe movement from vacuoles to the cytoplasm, which in turn alleviates the growth inhibition in an Fe-deficient environment.

5) Line 135-136 suggests GmNRAMP2a&2b are Fe influx transporters in yeast. Is this influx into vacuoles? If so can a Fe distribution analysis be completed in yeast cells expressing NRAMP2b to confirm this function?

Answer: We apologize for not having clearly conveyed the intended meaning of our statement. What we aim to express is that GmNRAMP2a and GmNRAMP2b mediate the transport of Fe from the vacuoles to the cytoplasm. We have made the necessary correction in our text.

6) Fig 5f highlights ultrastructural cell changes with \pm HN in WT and nramp2ab mutants. Its not clear if this is a Fe intensity image or not as the legend doesn't explain 5f.

Answer: Yes, it is a Fe intensity image. We apologize for the omission of essential details in the figure legend. We have now revised the legend to include all the necessary information.

7) Fig 5h/5n. I don't understand the comment that the nramp2a2b double mutant has a lower number of infected cells having lytic vacuoles re-appearing in the cells. The image would suggest the opposite to the quantified data?

Answer: Thank you for the questions. Figure 5h compares the wild-type with the nramp2ab mutants, whereas Figure 5n compares the wild-type with NRAMP2ab overexpression lines. To prevent confusion between the images, we have edited the figures, using distinctly different colors to differentiate between the mutants and overexpression lines.

Reviewer #3 (Remarks to the Author):

This paper shows that in the soybean-rhizobium symbiosis, inhibition of iron transport via NRAMP2a/b is associated with inhibition of symbiotic nitrogen fixation by externally supplied nitrogen nutrition. The authors show that NRAMP2 is localized in non-infected cells and regulates iron transport to infected cells, and that double mutants of NRAMP2a/b have abnormal iron transport to infected cells, resulting in reduced nitrogen fixation. In addition, they show that a GARP-type transcription factor, NIGT1, is responsible for the repression of NRAMP2a/b expression by nitrogen nutrition. The manuscript is generally well-written, and the data showing the function of NRSAMP2 is compelling. However, several points require further validation.

Thank you very much for your positive comments.

Comments:

Nitrate and ammonia are both known to inhibit root nodule symbiosis, but they are thought to act in different ways. Although NH_4NO_3 is used as the N source in this paper, whether nitrate or ammonia is actually more effective in controlling NRAMP2a/b should be examined for several important

results.

Answer: Thank you for your suggestion. We have conducted a time-course analysis of the expression response of NRAMP2ab to different nitrogen sources (ammonium and nitrate), and we found that both sources significantly inhibit the expression of *NRAMP2ab*, albeit with varying degrees of inhibition, yet the general pattern remains consistent (data were shown in figures below, Supplementary Fig. 2a-b). Previous studies have shown the upstream transcription factor NIGT1 exhibits a stronger response to nitrate than ammonium (Sawaki et al., 2013; Maeda et al., 2018; Kiba et al., 2018), which may be responsible for NRAMP2ab's reaction to nitrate. However, how NRAMP2ab respond to ammonium is unknown. Our finding indicate that although both ammonium and nitrate can inhibit nitrogen fixation in nodules, the mechanisms regulating this process might be distinct. We have added these results to the manuscript.

References:

- Sawaki, N. et al. A nitrate-inducible GARP family gene encodes an auto-repressible transcriptional repressor in rice. *Plant Cell Physiol.* 54, 506-517 (2013).
- Kiba, T. et al. Repression of Nitrogen Starvation Responses by Members of the Arabidopsis GARP-Type Transcription Factor NIGT1/HRS1 Subfamily. *Plant Cell* 30, 925-945 (2018).
- Maeda, Y. et al. A NIGT1-centred transcriptional cascade regulates nitrate signaling and incorporates phosphorus starvation signals in Arabidopsis. *Nat. Commun.* 9, 1376 (2018).

I do not agree with L172-174 and 182-183. The mutants appear to be affected by H-N to the same

extent as WT.

Answer: Thank you for your feedback. We have quantified the degree to which the wild-type and mutants are inhibited by high nitrogen levels, and updated the descriptions in the text: We observed 38% and 35% inhibitions by H-N in N export rate in the WT and the mutants, respectively. Meanwhile, the WT showed a 33% inhibition by H-N in Fe accumulation, whereas the mutants displayed a lower inhibition of 16% (Fig. 5e-g).

Fig. 3k: It is difficult to understand the effect of H-N on protein localization; why not examine vascular bundles where NRAMP2 is highly localized?

Answer: Thank you for the suggestion. We have also analyzed the impact of high nitrogen on the abundance of the NRAMP2ab proteins in the vascular bundles and found results consistent with those from the nitrogen-fixation regions; high nitrogen levels lead to a significant reduction in the abundance of these proteins (results are shown in the figure below, Supplementary Fig. 5e-f).

As high nitrogen significantly inhibits the expression of the *NRAMP2* genes, the aim of our experiment was to clarify whether high nitrogen also affects the abundance of these proteins. However, we do not exclude the possibility that protein levels may also be regulated by high nitrogen (an issue we did not address in this paper). How high nitrogen affects NRAMP2 protein levels is one of the future directions in our research. We hope to answer this question in our future

studies. Thank you for your understanding.

Data showing regulation of NRAMP2 by NIGT1 are scarce and not convincing; the relationship between the interaction and gene expression *in vivo* and *in vitro* should be investigated. Functional analysis of NIGT1 is also insufficient. As with NRAMP2, analysis needs to be performed using stable knockout plants.

Answer: Thank you for your valuable suggestion. We have performed an Electrophoretic Mobility Shift Assay (EMSA) experiment to investigate their interaction *in vitro* (results were shown below, Fig. 6h). We found that both GmNIGT1a and 1b could bind to all five of cis elements, although their binding affinity to P5 was markedly weaker. We reason that the adjacent upstream and downstream sequences, as well as their respective positions, may also influence the protein's binding ability.

For *in vivo* interactions, we have investigated the direct binding effect of NIGT1a and 1b on the upstream promoter of NRAMP2 utilizing the yeast system (Fig. 6d). Simultaneously, a six-fold

enrichment of the cis element (6xNIE) was constructed and demonstrated to be directly bound by NIGT1a and 1b (Fig. 6d). These results suggest the abilities of NIGT1a and 1b to directly bind to the NRAMP2 promoter *in vivo*.

For the functional analysis of NIGT1, we are fortunate to receive the *GmNIGT1ab* double knockout line (*nigt1ab-cr*) from Prof. Yuefeng Guan (Guangzhou University, China). The phenotype of the double knockout line was similar to what we previously observed in the RNAi lines: Although there were no differences in plant growth, nodule weight, and number, there was a significant effect on the gene expression of *NRAMP2*, the N export rate, and the accumulation and distribution of Fe in nodules (data were shown in figures below, Fig. 7). We have added these new results into the revised version.

Besides, to explore more downstream targets of GmNIGT1a&1b, we have conducted a comparative transcriptomic assay, which revealed 47 potentially NIGT1ab-regulated genes (as shown in Supplemental Dataset 3). These genes are up-regulated in *NIGT1ab* double knockout samples and down-regulated in *NIGT1a*-overexpressing materials and under high-nitrogen (H-N) conditions. Interestingly, we found that, beyond *NRAMP2ab*, important iron-homeostasis-related genes such as *IMAs* and *BRUTUS*s are also under the regulation of NIGT1ab (as shown in the figure below, Supplementary Fig. 17), which further strengthen our viewpoint regarding the regulation of Fe homeostasis by NIGT1 in nodules. We hope that these new findings will enhance our understanding

of the functions of NIGT.

There may be the impression that the statistics are done arbitrarily. In some graphs, only the results of a comparison of two conditions are shown, even though multiple data are shown in the same graph.

Answer: Thank you for your careful checking. Indeed, we have made comparisons between all conditions, but only displayed results where there were significant differences. In the revised version, we have re-conducted statistical analyses for all the figures in the text, and differential (asterisk) and non-differential (ns) annotations were shown.

The overexpression experiment uses the 35S promoter, yet no explanation is given for the fact that the effects of overexpression are nodule-specific.

Answer: Thank you for your suggestion. The 35S promoter has been previously reported to be primarily expressed in uninfected tissues of nodules (Auriac and Timmers, 2007), and our intention was to use this promoter to allow for overexpression of the target genes at least in uninfected cells. However, we acknowledge that 35S-driven overexpression is not nodule-specific, and its expression in roots and shoots may cause indirect effects on nodule development. We had planned to use nodule-specific promoters for overexpression, but to date, no promoter has been reported to be specific to uninfected cells in soybean nodules. To address this, we attempted to use the native promoter to drive *GmNRAMP2b* expression. But due to its relatively weak expression, we did not

observe any differences from the wild type (data not shown).

Alternatively, we used the *GmNIGT1a* promoter to drive *GmNRAMP2b* and the *GmNRAMP2b* promoter to drive *GmNIGT1a* expression in hairy roots (results shown as below, Supplementary Fig 16), to mitigate the regulation of genes by high nitrogen levels. As expected, nodules with *pGmNIGT1a:GmNRAMP2b-GFP* exhibited a stronger GFP signal under high nitrogen conditions, and the Fe accumulation and distribution in these nodules followed a trend similar to nodules with *p35S:GmNRAMP2b*. Nodules containing *pGmNRAMP2b:GmNIGT1a-GFP* showed a stronger GFP signal in control conditions than in high nitrogen treatment, and the Fe accumulation and distribution under control conditions mirrored the trend in nodules with *p35S:GmNIGT1a*. Therefore, we believe that the *GmNIGT1-GmNRAMP2* module specifically and directly regulates iron homeostasis in nodules. Contrary to our expectations, we did not observe any difference in number or weight of nodules between these transgenics and wild type nodules (data not shown). This could be due to considerable biological variation among the transgenic hairy roots or potentially indirect effects from using the 35S overexpression lines. To avoid misleading readers, we decide not to present data on nodule number and weight, and we appreciate your understanding.

Reference:

Auriac, M.C. & Timmers, A.C.J. Nodulation Studies in the Model Legume *Medicago truncatula*: Advantages of Using the Constitutive EF1 α Promoter and Limitations in Detecting Fluorescent Reporter Proteins in Nodule Tissues. *Mol. Plant-Microbe Interact.* 20, 1040-1047 (2007).

In each analysis, it is sometimes unknown whether stable transformants or hairy roots are used.

Answer: Sorry for the lack of clarity. In this study, most of the materials used were stable transgenic lines, as listed below: *nramp2b* (CRISPR-Cas9 editing), *nramp2ab* (CRISPR-Cas9 editing), *GmNRAMP2a-OE* (overexpression), *GmNRAMP2a&2b-OE* (overexpression), *nigt1ab-cr* (CRISPR-Cas9 editing), *nigt1ab-RNAi* (knockdown), and *GmNIGT1a-OE* (overexpression). We have provided descriptions for each material in the revised text. The phenotypic statistics for the overexpression and RNAi materials were based on two independent lines, and for each treatment, at least three plants per line were evaluated.

REVIEWER COMMENTS

Reviewer #1 (Remarks to the Author):

Review of resubmission: Inorganic nitrogen inhibits symbiotic nitrogen fixation through blocking NRAMP2-mediated iron delivery in soybean nodules.

The authors have addressed all my comments and this and the other changes have made the results more convincing. The proposed role of NRAMP2a and b is really interesting and provides another twist to the regulation of iron supply in the symbiosis.

I have a few suggestions on minor issues that could be addressed.

Sup figure 1b: figure legend says similar amino acids are in grey but many are in black. There is something a bit odd about the text showing the consensus

L91-92: “Real-time RT-PCR showed that both genes are primarily expressed in nodules, particularly in the fixation zone (Fig. 2e).” The real-time data is only for nodules and root (does nodule conjugated root mean those roots also have nodules on them?) so it does not show the genes are primarily expressed in nodules. NRAMP2a is not primarily expressed in nodules (see 2d). Needs correction.

The expression of the genes in figure 2f of the resubmitted manuscript and quite different to those in 2f of the old one. Is this the difference between hydroponic and vermiculite grown plants?

Figure legend for Fig 2. Part d shows that expression is not nodule specific so change 2e to “Expression in nodules” or similar. “nodule conjugated root segments” - does this mean there were nodules on the roots or were they roots with nodules removed. Make this clear. Given expression in figure the nodules may have been removed. f. some text missing in days after inoculation on graph. g. there is no key for the colours in the figure.

Sup Figure 3 legend: Plants not nodules were transplanted. In (a-d) and (g-h) Change “Transgenic nodules at 23 dpi from hairy roots weretransplanted to Fe-free” to “Plants (or transgenic lines) with

hairy roots and nodules 23 dpi were transplanted to Fe-free..." (g-h) delete "staining". Same in Sup figure 5b.

Sup figure 10 – text is a bit small, particularly that below heat maps. The red labels for particular genes are hard to match to the rows in the heat map.

Sup fig10- Text below mentions nramp2b-1, nramp2b-2 – should this be nramp2ab-1, nramp2ab-2 as in figure legend.

Sup figure 10 : The figure legend is not clear about what the boxes with arrows to the left of the heatmaps mean.

Lines 189-203: This was added to describe the results from Sup figures 10 and 11 and to respond to a point by reviewer 2. I didn't find the text or the figures particularly clear to address the point made. Sup figure 10 presents a lot of data but most was not mentioned in the text.

Points to address:

1. While I can see the transcripts of some ferritin genes are reduced in nrampab mutants, the western blot for ferritin doesn't show an obvious decrease in nramp2ab mutants so is it actually indicating intracellular Fe levels?

2. The figure legends (particularly sup fig 10) needs to be made clearer.

3. I probably agree with the conclusion that "These results suggest that Fe homeostasis in nodules relies on the complex interplay and coordination of multiple transporters" but wonder at the value of the extra figures. Could the data be compressed to make a clear point.

Line208: Edit to make clearer- I suggest: Meanwhile, the WT showed a 33% inhibition (should this be reduction?) in Fe accumulation in H-N, whereas the mutants displayed only 16% reduction

Lines 264-265: What promoters were used for the RNAi (nigt1ab-RNAi) and OE (GmNIGT1a-OE). Could this be added to the text or the figure legend.

Line278, 280: correct GmNRMAP2b

Line 304: delete "all"

Line 317: The implication of them as influx transporters that localise on the vacuole in planta should be made clear ie. They would transport Fe from the vacuole into the cytoplasm

Line 328: release OF Fe

Line 330: delete “level of”. Change “it is not as severe as in vtl1 mutants” to “the phenotype of the mutant is not as severe as for vtl1 mutants”

Line332: correct “invovled”

Line 334: change to : responsible for transport of Fe from the apoplastic into infected cells. The equivalent transporter in soybean may collaborate with NRAMP2ab to ensure....”

Line 360: transducerS

Reviewer #2 (Remarks to the Author):

The manuscript has dramatically improved with the addition of key technical detail as requested. I'm not sure why this wasn't presented in the first place. The re-aligned materials and methods read better and aligns with the data presented.

The inclusion of further Fe-related experiments using both nramp2a/b and vtl1 mutants does help to better understand possible Fe homeostatic controls in nodules. However, I'm a bit perplexed with some of the data. The ferritin protein analysis suggests there is no signal in the control nodules. I would suspect these cells to be Fe sufficient and emulate previous ferritin detection in nodules (Chikoti et al 202 JPP). If nramp2a or 2b have a negative impact on N₂-fixation capacity, then why does NifH remain stable while lost in the vtl1 mutants. I agree with the possible explanation that multiple processes are operating to manage Fe availability. The manuscripts overall pitch should reflect this complimentary role of the two NRAMP2 and NIGT proteins.

The assay used to define intercellular localisation of NRAMP2a/b with a C-terminal GFP tag is very convincing, though the extra controls in this rebuttal still fail to conclusively refute the previous observations made using GmNRAMP2b specific antibodies with immunogold labelling and western

blot analysis. Differences could be due to the stability of transcripts in infected cells as the clone used in this study contained a PCR amplified open reading frame, while previous studies had looked at tissues expressing native transcripts which retain both 5' and 3' untranslated regions, with the 3'- retaining the terminal IRE motif, a known Fe responsive element. I suggest to conclusively show where these proteins are located, the authors should generate similar peptides as previously used and raise antibodies followed by new immunogold and western blot analysis. The authors did raise a Nod26 antibody and detected native protein using western blots. A full-length genomic clone could be fused as was done in the NITG experiments. Furthermore, a faint GFP fluorescence still resides inside infected cells (Figures 3D and particularly 3J when +Fe is used). NRAMP2a/2b are most likely strongly expressed in uninfected cells, but the data fails to rule out other cell types including infected cells. The text should reflect this.

I appreciate the completion of further Fe uptake experiments in yeast. There appears to be inward flux of ^{57}Fe (10 μM) by GmNRAMP2b. However, I expect the sensitivity of the stable isotope (^{57}Fe) assays at low concentrations (<10 μM) limit their usefulness to define net transport capabilities. At 10 or 100 μM , Fe concentrations are very high for yeast cells and easily complement mutants such as Fet3Fet4. A proper characterisation should be done using short-term (5 min) ^{55}Fe influx assays if the authors want to characterise the transport properties of their respective clones. I would also suggest full-length cDNA's be expressed in yeast cells rather than open reading frame cDNAs to be consistent with previously published work.

I can see the localisation of GmNRAMP2a/b to the tonoplast in tobacco cells. However, we haven't seen that targeting in yeast cells in this data set? This is important to confirm the results presented in Fig 4b. To confirm the direction of Fe transport (into or out of the vacuole) the assays should be set-up with an inducible yeast promoter to initiate time-dependent leakage from vacuoles. The authors have used pYES3, which retains a Gal promoter, this would be suitable for this type of work.

Reviewer #3 (Remarks to the Author):

The revised manuscript has mostly addressed my concerns, reinforcing the author's claim. An additional minor comment as follows.

Similar to GmNRAMP2a/b genes, Four GmIMA peptide genes are respectively downregulated and upregulated in H-N and -Fe conditions in their transcriptome analysis (Fig. 2c), but the authors did not explain the GmIMA genes expression. Recently, Lotus IMA peptides have been reported to regulate nitrogen fixation (Ito et al. Nat Commun, 2024). This paper should be cited and briefly discuss their relation to the function of GmNRAMP2a/b.

REVIEWER COMMENT

Reviewer #1 (Remarks to the Author):

Review of resubmission: Inorganic nitrogen inhibits symbiotic nitrogen fixation through blocking NRAMP2-mediated iron delivery in soybean nodules.

The authors have addressed all my comments and this and the other changes have made the results more convincing. The proposed role of NRAMP2a and b is really interesting and provides another twist to the regulation of iron supply in the symbiosis.

Answer: Thank you again for your comments.

I have a few suggestions on minor issues that could be addressed.

Sup figure 1b: figure legend says similar amino acids are in grey but many are in black. There is something a bit odd about the text showing the consensus.

Answer: Sorry for the unclear description. We have rephrased this sentence: Amino acid sequences identical in all three are highlighted in black, and those with two identical sequences are highlighted in grey.

L91-92: “Real-time RT-PCR showed that both genes are primarily expressed in nodules, particularly in the fixation zone (Fig. 2e).” The real-time data is only for nodules and root (does nodule conjugated root mean those roots also have nodules on them?) so it does not show the genes are primarily expressed in nodules. NRAMP2a is not primarily expressed in nodules (see 2d). Needs correction.

Answer: Yes. Nodule conjugated root means those roots without nodules on them. We have annotated this in the revised text. We have also corrected this sentence: Real-time RT-PCR showed that *GmNRAMP2b* are primarily expressed in the fixation zone of nodules (Fig. 2e).

The expression of the genes in figure 2f of the resubmitted manuscript and quite different to those in 2f of the old one. Is this the difference between hydroponic and vermiculite grown plants?

Answer: Yes. We reason that this expression difference is mainly due to the growth medium between the two experiments, which leads to the differences in seedling growth stage and nodule maturity. For the old one, 4-d-old seedlings were inoculated with rhizobia and grown in a **vermiculate** medium supplied with low-N solution for 7, 14, 21, 30, and 50 days. Considering that all experimental conditions in the paper were conducted under hydroponic conditions, we therefore rechecked the time-course expression pattern using seedlings grown under **hydroponic culture**. The results from the hydroponically grown seedlings indicate that the expression of *NRAMP2b* is much earlier to reach the peak than that grown in vermiculate. We speculate that under hydroponic conditions (6 seedlings in 12 L container), the root system has sufficient space and nutrients to achieve its optimal growth state, compared to that grown in vermiculite (3 seedlings in 1.5 L pot).

Figure legend for Fig 2. Part d shows that expression is not nodule specific so change 2e to “Expression in nodules” or similar. “nodule conjugated root segments”- does this mean there were nodules on the roots or were they roots with nodules removed. Make this clear. Given expression I figure the nodules may have been removed. f. some text missing in days after inoculation on graph. g. there is no key for the colours in the figure.

Answer: Thank you for your careful checking. We have changed the title of Fig. 2e into “Gene expression in nodules”. “nodule conjugated root segments” are roots without nodules, we have clarified in the revised text. We have corrected the Fig. 2f and 2g accordingly.

Sup Figure 3 legend: Plants not nodules were transplanted. In (a-d) and (g-h) Change “Transgenic nodules at 23 dpi from hairy roots were transplanted to Fe-free” to “Plants (or transgenic lines) with hairy roots and nodules 23 dpi were transplanted to Fe-free...” (g-h) delete “staining”. Same in Sup figure 5b.

Answer: Thanks. We have changed these words accordingly.

Sup figure 10 – text is a bit small, particularly that below heat maps. The red labels for particular genes are hard to match to the rows in the heat map.

Answer: Thank you for your suggestion. We have adjusted the font and bolded it. The highlighted genes with a red color have been repositioned to match the rows.

Sup fig10- Text below mentions nramp2b-1, nramp2b-2 – should this be nramp2ab-1, nramp2ab-2 as in figure legend.

Answer: Sorry for this mistake. *nramp2b* has been corrected to *nramp2ab*.

Sup figure 10 : The figure legend is not clear about what the boxes with arrows to the left of the heatmaps mean.

Answer: We have added the information to the figure legend: The arrows in the left box of the heatmap indicate whether the gene is upregulated (upward) or downregulated (downward) in the mutant (*nramp2ab* or *vt11*), in comparison to the WT.

Lines 189-203: This was added to describe the results from Sup figures 10 and 11 and to respond to a point by reviewer 2. I didn't find the text or the figures particularly clear to address the point made. Sup figure 10 presents a lot of data but most was not mentioned in the text.

Answer: Thanks for your suggestions. We realized that the data in Supplementary Fig. 10 is already presented in Supplementary Dataset 2. Therefore, to present these differential data more succinctly, we have simplified Supplementary Fig. 10 and 11, by: 1. removing the -Fe and H-N treatment data, as the trends are similar to CK; 2. deleting genes that do not show differences between the wild type and the mutants. On the other hand, we have revised the text, providing more detailed descriptions of Supplementary Fig. 10. See L202-207.

Points to address:

1. While I can see the transcripts of some ferritin genes are reduced in nrampab mutants, the western blot for ferritin doesn't show an obvious decrease in nramp2ab mutants so is it actually indicating intracellular Fe levels?

Answer: Thanks for your suggestions. Due to the very high levels of ferritin in the *vll* mutants, we kept the western blot exposure time very short to prevent overexposure. This resulted in the ferritin bands being almost invisible in the wild type under short exposure conditions. We apologize for any confusion this may have caused. We have updated these images by extending the exposure time. We can see that the ferritin levels in the wild type were higher than those in the *nramp2ab* mutants, although the protein levels of ferritin in nodules are indeed not high (as shown in the figure below).

2. The figure legends (particularly sup fig 10) needs to be made clearer.

Answer: Thanks for your suggestions. We have rechecked all the figure legends and made some revisions to enhance clarity.

3. I probably agree with the conclusion that “These results suggest that Fe homeostasis in nodules relies on the complex interplay and coordination of multiple transporters” but wonder at the value of the extra figures. Could the data be compressed to make a clear point.

Answer: Thanks for your suggestions. The purpose of presenting this data is to illustrate the point that mutations in either *NRAMP2* or *vll* can trigger changes in many other iron transporters within the nodules. This response by host plant aims to compensate for the defects in nodule Fe supply caused by these gene mutations. We realized that the data in Supp Fig. 10 is already presented in Supp Dataset 2. Therefore, to present these differential data more succinctly, we have simplified Supp Fig. 10 by: 1. removing the -Fe and H-N treatment data, as the trends are similar to CK; 2. deleting genes that do not show differences between the wild type and the mutants. We hope the simplified figure will be easier to read and understand.

Line208: Edit to make clearer- I suggest: Meanwhile, the WT showed a 33% inhibition (should this be reduction?) in Fe accumulation in H-N, whereas the mutants displayed only 16% reduction

Answer: We have revised accordingly.

Lines 264-265: What promoters were used for the RNAi (nigt1ab-RNAi) and OE (GmNIGT1a-OE). Could this be added to the text or the figure legend.

Answer: We used 35S promoter. We have added this information into the text.

Line278, 280: correct GmNRMAP2b

Answer: Thanks for the careful checking. We have corrected accordingly.

Line 304: delete “all”

Answer: We have deleted it accordingly.

Line 317: The implication of them as influx transporters that localise on the vacuole in planta should be made clear ie. They would transport Fe from the vacuole into the cytoplasm

Answer: We have revised accordingly.

Line 328: release OF Fe

Answer: We have revised accordingly.

Line 330: delete “level of”. Change “it is not as severe as in vt11 mutants” to “the phenotype of the mutant is not as severe as for vt11 mutants”

Answer: We have revised accordingly.

Line332: correct “invovled”

Answer: Thanks for the careful checking. We have corrected accordingly.

Line 334: change to : responsible for transport of Fe from the apoplastic into infected cells. The equivalent transporter in soybean may collaborate with NRAMP2ab to ensure....”

Answer: We have revised accordingly.

Line 360: transducerS

Answer: We have corrected accordingly. Thank you again for these helpful comments.

Reviewer #2 (Remarks to the Author):

The manuscript has dramatically improved with the addition of key technical detail as requested. I'm not sure why this wasn't presented in the first place. The re-aligned materials and methods read better and aligns with the data presented.

The inclusion of further Fe-related experiments using both *nramp2a/b* and *vt11* mutants does help to better understand possible Fe homeostatic controls in nodules. However, I'm a bit perplexed with some of the data. The ferritin protein analysis suggests there is no signal in the control nodules. I would suspect these cells to be Fe sufficient and emulate previous ferritin detection in nodules (Chikoti et al 202 JPP). If *nramp2a* or *2b* have a negative impact on N₂-fixation capacity, then why does NifH remain stable while lost in the *vt11* mutants. I agree with the possible explanation that multiple processes are operating to manage Fe availability. The manuscripts overall pitch should reflect this complimentary role of the two NRAMP2 and NIGT proteins.

Answer: Thank you again for your suggestions. Due to the very high levels of ferritin in the *vt11* mutants, we kept the western blot exposure time very short to prevent overexposure. This resulted in the ferritin bands being almost invisible in the wild type under short exposure conditions. We apologize for any confusion this may have caused. We have updated these images by extending the exposure time, and treating the different mutants separately. We can see that the ferritin levels in the wild type were higher than those in the *nramp2ab* mutants, although the protein levels of ferritin in nodules are indeed not high (as shown in the figure below). This trend aligns with the transcriptomic data (Supplementary Fig. 10), providing a more precise quantification of ferritin expression.

In terms of NifH protein abundance, although the decrease in *nramp2ab* is not as pronounced as in *vtl1*, there is still a significant reduction. We have quantified the ratio of NifH to Actin (as shown in the figure below). In old version of this manuscript, we also provided quantified results, demonstrating a noticeable decrease in NifH to H⁺-ATPase ratio in *nramp2ab* compared to WT (as shown in the figure below). We reason that the NifH differences between *nramp2ab* and *vtl* mutants is because they play different roles in soybean nodules. VTL1 is an indispensable protein for maintaining the basic N fixation of infected cells, while NRAMP2 predominantly functions as a regulator in uninfected cells, with its role becoming more pronounced when the nodule has a higher demand for Fe (see L330-338).

Additionally, we have made modifications to the manuscript, including the results and discussion sections, to clearly highlight the complimentary role of NRAMP2ab and NIGT1ab proteins in nodules. For NRAMP2a and 2b, we also highlighted the presence of genetic compensation between them. See L345-358.

The assay used to define intercellular localisation of NRAMP2a/b with a C-terminal GFP tag is very convincing, though the extra controls in this rebuttal still fail to conclusively refute the previous observations made using GmNRAMP2b specific antibodies with immunogold labelling and western blot analysis. Differences could be due to the stability of transcripts in infected cells as the clone used in this study contained a PCR amplified open reading frame, while previous studies had looked at tissues expressing native transcripts which retain both 5' and 3'untranslated regions, with the 3'- retaining the terminal IRE motif, a known Fe responsive element. I suggest to conclusively show where these proteins are located, the authors should generate similar peptides

as previously used and raise antibodies followed by new immunogold and western blot analysis. The authors did raise a Nod26 antibody and detected native protein using western blots. A full-length genomic clone could be fused as was done in the NITG experiments. Furthermore, a faint GFP fluorescence still resides inside infected cells (Figures 3D and particularly 3J when +Fe is used). NRAMP2a/2b are most likely strongly expressed in uninfected cells, but the data fails to rule out other cell types including infected cells. The text should reflect this.

Answer: Thank you for these constructive suggestions. It has come to our attention that we overlooked the impact of 5' and 3' untranslated regions on gene expression and protein localization. Our choice of promoter initiates transcription before the start codon ATG, encompassing the 5' UTR, though the 3' UTR was not included. To address the role of the 3' UTR, we reconstructed the vector proGmNRAMP2b: GmNRAMP2b-GFP-3' UTR and investigated transgenic nodules. The result showed that anti-GFP signals were still mainly localized in uninfected cells (as shown in the figure below). This indicates that the 3' UTR at least do not influence the pattern and distribution of the GmNRAMP2b protein between uninfected and infected cells.

Furthermore, we utilized the vector proGmNRAMP2b: gGmNRAMP2b-GFP to assess the impact of the full-length genomic sequence (including introns) on protein distribution. Similar to the ORF, our results indicate that gGmNRAMP2b is still localized on the tonoplast of uninfected cells (as shown in the figure below).

Regarding the preparation of the GmNRAMP2b-specific antibodies, this process began at the early stage of this project. We utilized both proteins 1-80 aa in length (following a method similar to Kaiser et al., 2003) and synthetic peptides from 498-512aa to immunize rabbits for obtaining polyclonal antibodies against GmNRAMP2b. Unfortunately, our investigations by both immunostaining and western blot in mutant and wild-type samples revealed that these antibodies lacked specificity (signals detected in WT samples also can be detected in mutants, data not shown). It is important to note that during polyclonal antibody preparation, nonspecific reactions can be common and unavoidable, especially when the antigen structure is similar to other molecules in the host. However, each antibody preparation is an independent event, and the lack of specificity in our assays does not imply that the antibodies used by Kaiser et al. 2003 are non-specific. Consequently, we decided to proceed with experiments using GFP tag instead, as its antibody is commercially available and shows high specificity. Thank you for your understanding.

Considering the possible expression of this protein in infected cells, we have addressed this in the discussion section of the manuscript: Nevertheless, *GmNRAMP2b/DMT1* was still expressed at a low level in infected cells (Fig. 3d-f), suggesting its potential minor role in these cells.

I appreciate the completion of further Fe uptake experiments in yeast. There appears to be inward flux of ^{57}Fe (10 μM) by GmNRAMP2b. However, I expect the sensitivity of the stable isotope (^{57}Fe) assays at low concentrations (<10 μM) limit their usefulness to define net transport capabilities. At 10 or 100 μM , Fe concentrations are very high for yeast cells and easily complement mutants such as Fet3Fet4. A proper characterisation should be done using short-term (5 min) ^{55}Fe influx assays if the authors want to characterise the transport properties of their respective clones. I would also suggest full-length cDNA's be expressed in yeast cells rather than

open reading frame cDNAs to be consistent with previously published work.

Answer: Thank you again for your suggestion. We understand that using the short-term radioactive isotope ^{55}Fe is the best way to identify iron influx of transporter. Unfortunately, we are unable to obtain ^{55}Fe and we do not have the qualifications to conduct radioactive isotope experiments. Therefore, we have chosen the stable isotope ^{57}Fe as an alternative. Although it does not have the advantage of no background contamination like ^{55}Fe , the natural abundance of ^{57}Fe is only 2.1%. The background levels detected in yeast are almost negligible compared to the amount of ^{57}Fe absorbed. Due to its convenience in handling and low natural abundance, ^{57}Fe has been used in many in vivo iron uptake studies, including those in yeast and plants (Shepherd et al., 2023; Che et al., 2021; Yamaji et al., 2024).

On the other hand, based on your suggestions, we conducted low concentration (1 and 5 μM) ^{57}Fe uptake experiments. The results showed that yeast expressing NRAMP2a and NRAMP2b still showed higher ^{57}Fe uptake than the control. This suggests that iron absorption by NRAMP2 in yeast is a direct transport process.

References:

Shepherd, R. E., Kreinbrink, A. C., Njimoh, C. L., Vali, S. W., & Lindahl, P. A. Yeast mitochondria import aqueous FeII and, when activated for iron–sulfur cluster assembly, export or release low-molecular-mass iron and also export iron that incorporates into cytosolic proteins. *J. Am. Chem. Soc.* 145, 25, 13556-13569 (2023).

Che, J., Yamaji, N. & Ma, J.F. Role of a vacuolar iron transporter OsVIT2 in the distribution of iron to rice grains. *New Phytol.* 230, 1049–1062 (2021).

Yamaji, N., Yoshioka, Y., Huang, S., Miyaji, T., Sasaki, A. & Ma, J.F. An oligo peptide transporter family member, OsOPT7, mediates xylem unloading of Fe for its preferential distribution in rice. *New Phytol.* 242, 2620–2634 (2024).

I can see the localisation of GmNRAMP2a/b to the tonoplast in tobacco cells. However, we haven't seen that targeting in yeast cells in this data set? This is important to confirm the results presented in Fig 4b. To confirm the direction of Fe transport (into or out of the vacuole) the assays should be set-up with an inducible yeast promoter to initiate time-dependent leakage from vacuoles. The authors have used pYES3, which retains a Gal promoter, this would be suitable for this type of work.

Answer: Thanks for your suggestions. We have investigated the protein localization in yeast cells, which was shown in Supplementary Fig. 4a (also see figure below). We found both of them localized specifically to vacuolar membrane (see text L118-119).

We appreciate your suggestions regarding our yeast experiments. In our initial experimental design, we cultured the yeast in glucose conditions until reaching the exponential phase, followed by transferring them to galactose and 1 mM FeSO₄ for 2 hours to induce gene expression. In the newly updated manuscript, we conducted new experiments with 0, 0.5, 1 and 2 hour of galactose induction (We apologize for not being able to investigate more time points, considering the heavy workload required for the extraction and collection of yeast vacuoles). This time, we have also included a transgenic yeast strain expressing the full-length cDNA of GmNRAMP2b as you suggested. Our results indicate that the iron accumulation in the yeast vacuoles increases with longer induction times. However, yeast cells carrying GmNRAMP2ab ORF or full-length cDNA exhibited lower iron accumulation in the vacuoles at the 1 and 2-hour time points (as shown in the figure below). The difference in iron accumulation at the 0.5-hour time point was not significant, possibly due to the insufficient gene expression levels induced by galactose for a short time.

Reviewer #3 (Remarks to the Author):

The revised manuscript has mostly addressed my concerns, reinforcing the author's claim. An additional minor comment as follows.

Similar to GmNRAMP2a/b genes, Four GmIMA peptide genes are respectively downregulated and upregulated in H-N and -Fe conditions in their transcriptome analysis (Fig. 2c), but the authors did not explain the GmIMA genes expression. Recently, Lotus IMA peptides have been reported to regulate nitrogen fixation (Ito et al. Nat Commun, 2024). This paper should be cited and briefly discuss their relation to the function of GmNRAMP2a/b.

Answer: Thank you again for your suggestions. We have added some discussion on the function of IMA in plants and nodules as follows: Meanwhile, the genes downstream of NIGT1 include not only *NRAMP2* but also *IMAs* and *BRUTUSs* (Supplementary Fig. 17b). IMA peptides positively regulate Fe homeostasis in plants by interacting with the E3 ubiquitin ligase BRUTUS, which is required for the degradation of transcription factors involved in the Fe deficiency response ^{49, 50}. In root nodules of *Lotus japonicus*, IMA peptides have recently been reported to regulate nitrogen fixation ⁵¹, which highlight the essential role of IMA-mediated Fe provision in regulating N-related physiological processes. Whether IMA peptides regulate NRAMP2 or act independently to maintain Fe balance in root nodules requires further study.

References:

49. Grillet, L., Lan, P., Li, W., Mokkapati, G. & Schmidt, W. IRON MAN is a ubiquitous family of peptides that control iron transport in plants. *Nat. Plants* **4**, 953-963 (2018).
50. Li, Y. et al. IRON MAN interacts with BRUTUS to maintain iron homeostasis in Arabidopsis. *Proc. Natl. Acad. Sci. USA*. **118**(39): e2109063118 (2021).
51. Ito, M. et al. IMA peptides regulate root nodulation and nitrogen homeostasis by providing iron according to internal nitrogen status. *Nat. Commun.* **15**(1):733 (2024).

REVIEWERS' COMMENTS

Reviewer #2 (Remarks to the Author):

I appreciate the extra experiments and rigour applied to the revised manuscript. The authors have addressed my concerns and presented a revised manuscript that reflects a substantial advance in understanding the regulation of Fe transport in legume nodules.